# Resident memory T cells are a cellular reservoir for HIV in the cervical mucosa

Jon Cantero-Pérez [1,10], Judith Grau-Expósito [1,10], Carla Serra-Peinado [1], Daniela A. Rosero [1], Laura Luque-Ballesteros [1], Antonio Astorga-Gamaza[1], Josep Castellví [2], Tamara Sanhueza[3], Gustavo Tapia[3], Belen Lloveras[4], Marco A. Fernández [5], Julia G. Prado [6], Josep M. Solé-Sedeno [7], Antoni Tarrats [8], Carla Lecumberri [8], Laura Mañalich-Barrachina[9], Cristina Centeno-Mediavilla[9], Vicenç Falcó [1], Maria J. Buzon [1]* & Meritxell Genescà [1]*

HIV viral reservoirs are established very early during infection. Resident memory T cells ($T_{RM}$) are present in tissues such as the lower female genital tract, but the contribution of this subset of cells to the pathogenesis and persistence of HIV remains unclear. Here, we show that cervical CD4$^+$$T_{RM}$ display a unique repertoire of clusters of differentiation, with enrichment of several molecules associated with HIV infection susceptibility, longevity and self-renewing capacities. These protein profiles are enriched in a fraction of CD4$^+$$T_{RM}$ expressing CD32. Cervical explant models show that CD4$^+$$T_{RM}$ preferentially support HIV infection and harbor more viral DNA and protein than non-$T_{RM}$. Importantly, cervical tissue from ART-suppressed HIV$^+$ women contain high levels of viral DNA and RNA, being the $T_{RM}$ fraction the principal contributor. These results recognize the lower female genital tract as an HIV sanctuary and identify CD4$^+$$T_{RM}$ as primary targets of HIV infection and viral persistence. Thus, strategies towards an HIV cure will need to consider $T_{RM}$ phenotypes, which are widely distributed in tissues.

[1] Infectious Diseases Department, Hospital Universitari Vall d'Hebron, Institut de Recerca (VHIR), Universitat Autònoma de Barcelona, Barcelona, Spain. [2] Pathology Department, Hospital Universitari Vall d'Hebron, UAB, Barcelona, Spain. [3] Pathology Department, Hospital Universitari Germans Trias i Pujol, Badalona, Barcelona, Spain. [4] Pathology Department, Hospital del Mar, Parc de Salut Mar, Universitat Autònoma de Barcelona, Barcelona, Spain. [5] Flow Cytometry Facility, Institut d'Investigació en Ciències de la Salut Germans Trias i Pujol, Badalona, Spain. [6] AIDS Research Institute IrsiCaixa, Institut d'Investigació en Ciències de la Salut Germans Trias i Pujol, Universitat Autònoma de Barcelona, Badalona, Spain. [7] Obstetrics and Gynecology Department, Hospital del Mar, Parc de Salut Mar, Universitat Autònoma de Barcelona, Barcelona, Spain. [8] Department of Obstetrics and Gynecology, Hospital Universitari Germans Trias i Pujol, Badalona, Barcelona, Spain. [9] Department of Obstetrics and Gynecology, Hospital Universitari Vall d'Hebron, Universitat Autònoma de Barcelona, Barcelona, Spain. [10] These authors contributed equally: Jon Cantero-Pérez, Judith Grau-Expósito. *email: mariajose.buzon@vhir.org; meritxell.genesca@vhir.org

The major hurdle to HIV-1 eradication is the establishment of reservoirs harboring latent HIV and low levels of viral replication while on antiretroviral therapy (ART)[1]. The characteristics of this reservoir and the viruses that emerge during ART interruption suggest that it is established in different tissues, including the mucosa[2], a few days after becoming infected. Early treatment with ART can prevent viremia and limit the size of the viral reservoir, but it does not prevent its establishment[1]. While the main HIV cellular target, activated CD4+ T cells, die massively during the initial infection, it is assumed that a very low proportion of the initially infected cells live long enough to be able to enter into a resting state (quiescence), establishing a memory reservoir. The fact that memory CD4+ T cells generate long-term protection (even during the whole life of the individual), without needing a second exposure to the pathogen and considering that its physiology includes the homeostatic proliferation of these cells, supports theories that only the long life of this population is sufficient for the maintenance of these reservoirs[2]. In addition, the presence of pharmacological and immunological sanctuary sites has been associated to residual viral replication in tissues, thus contributing to viral persistence[3,4].

It has been determined that in blood the virus persists mostly in the central memory and, to a lesser extent, in the transitional memory (CD27+) subsets[5]. Within the memory compartment, CD4+ T follicular helper cells recently emerged as an important population that sustains HIV reservoirs[6]. Likewise, the contribution to the total HIV reservoir of the stem cell-like memory T cells (T$_{SCM}$), defined mainly by the expression of CD95 in a population phenotyped as naive, increases significantly after long-term therapy, compared to other memory subpopulations[7]. Further, CD4+ T cells with high expression of CD2 have been postulated as potentially associated with latently infected cells[8]. More recently, CD32 expression in a subset of CD4+ T cells was also associated to HIV-latently infected cells[9], but soon after it was demonstrated to be linked to transcriptionally active infected cells[10–13]. Similarly, the effector memory phenotype and the expression of CD30 have also been associated to transcriptionally active infected cells[14,15]. Nevertheless, most of these populations have been determined in blood, while in tissues is where the main reservoir resides. In this sense, assessment of viral DNA (vDNA) and viral RNA (vRNA) positive cells in effectively ART-treated macaques (plasma vRNA < 50 copies/mL) has shown virus remaining in every tissue examined[3]. Further, in ART-suppressed patients proviral DNA determined in cells obtained from ileum biopsies[16] or from bronchoalveolar lavage[17] is enriched compared to blood, and even CD4+ T cells from adipose tissue represent a niche of viral persistence[18]. In macaque models these reservoirs are established within three days after infection[19] and therefore target cells present in the initially infected tissues may have great relevance in the generation and persistence of these reservoirs. Thus, tissues from the female genital mucosa, the portal of entry of HIV, may represent an unrecognized reservoir, as suggested[20,21]. In this sense, the most abundant hematopoietic cells in the cervix are CD14+/CD68+ cells, compatible with different myeloid phenotypes, and CD4+ T cells[22,23] likely belonging to the resident memory T cell (T$_{RM}$) phenotype, according to their CD69 expression[24].

CD4+ T$_{RM}$ reside in various tissues in mice and humans and some of their phenotypic characteristics have recently been established[25,26]. Since CD69 is a hallmark of these cells and, in general, the majority of cervical CD4+ T cells express this activation marker[24], this would question the role for these cells in the establishment of viral latency, as presumably they do not represent quiescent cells. However, the expression of this molecule in T$_{RM}$ may have a role in tissue retention, since it regulates the sphingosine-1 phosphate receptor, which in turn participates in the exit of lymphocytes to tissues[25,26]. Further, in T$_{RM}$ from lung or spleen, CD69 expression is not associated with expression of activation markers, namely CD25, CD38 and HLA-DR. This phenotype, together with a reduced expression of the proliferation marker Ki67 and maintenance of expression of CD127 and CD28, suggests that T$_{RM}$ may have a low turnover and a more quiescent state, which could prevent inadequate activation in tissues with high antigenic exposure while promoting the longevity of these cells[25].

In the macaque model of SIV infection, gut CD69+CD4+ T cells, which are predominant in this tissue, are the main viral target that is rapidly depleted by direct infection[27]. Furthermore, it has been shown that HIV entry in cytobrush-derived cervical CD4+ T cells of HIV− women is preferential in activated cells expressing CD69, α4β7, and α4β1, and its enhanced susceptibility strongly correlates with increased CCR5 expression[28]. While these studies highlight tissue CD69+CD4+ T cells as preferential targets of acute HIV infection, it remains to be elucidated if these cells belong to the T$_{RM}$ phenotype and if they encompass a long-term reservoir during ART. Thus, although CD69 expression on T$_{RM}$ may have precluded their study in relation to HIV-1 latency, several of their features suggest that these cells could contribute to HIV persistence: HIV-1 susceptibility, wide tissue distribution (including mucosal portals of entry for HIV) and local proliferation and longevity of different antigen specificities[25,29]. Here, to resolve if CD4+ T$_{RM}$ embody a cellular HIV-1 reservoir, we focus on cervical tissue, which we also postulate may represent an unrecognized HIV sanctuary. After detailed analysis of the CD4+ T$_{RM}$ protein profile in these mucosal tissues, which predicted increased susceptibility and long-term persistence, we show preferential infection of these cells using a cervical ex vivo model of HIV infection. To address if CD4+ T$_{RM}$ are indeed a source of HIV persistence, we studied cervical tissues from aviremic ART-treated HIV-1 infected women. Critically, we detect higher vDNA content in cervical tissues compared to contemporaneous blood samples from these patients, and show that CD4+ T$_{RM}$ harboring vDNA and vRNA are the main contributors to this reservoir.

## Results

**Evaluation of T$_{RM}$ cell markers in cervical CD69+ cells.** In recent extensive T$_{RM}$ analyses of human tissues[25,26], the female genital tract has not been included and, thus, the precise characteristics of cervical T$_{RM}$ remain largely undefined. Still, in the female reproductive tract of mice, a recent publication demonstrates that >90% of the CD4+ T cells are essentially T$_{RM}$[30]. In order to confirm that CD69 expression in these human mucosal tissues defines cells compatible with a T$_{RM}$ profile reported by others[25,31,32], we first analyzed the expression of several transcriptional factors and surface proteins previously associated to T$_{RM}$ in a small group of samples. The gating strategy used for these analyses and examples of the expression of these markers in the CD69− or the CD69+ fraction of CD4+ T cells derived from cervical tissues are shown in Fig. 1a, b, respectively. As reported for human and mouse T$_{RM}$[32], Eomes and T-bet were mostly absent from cervical CD69+ CD4+ T cells, while a median of 1.51% and of 3.5% CD69−CD4+ T cells expressed Eomes and T-bet, respectively (Fig. 1c, $p = 0.0625$ for T-bet). As recently reported for CD69+ CD8+ T cells derived from other human tissues[25], Hobit was minimally expressed in the CD69− fraction (median of 1.71%), but absent from the CD69+ CD4+ T cells ($p = 0.0625$). S1PR1 and CCR7, molecules expected to be downregulated in T$_{RM}$[25,31,32], were only detected in 3.18% and 3.05% of the CD69+ CD4+ T cells, respectively (Fig. 1c, $p = 0.0156$ for CCR7). In addition, CD49a and PD-1, the expression of which has also been associated to the core signature of T$_{RM}$[25], were

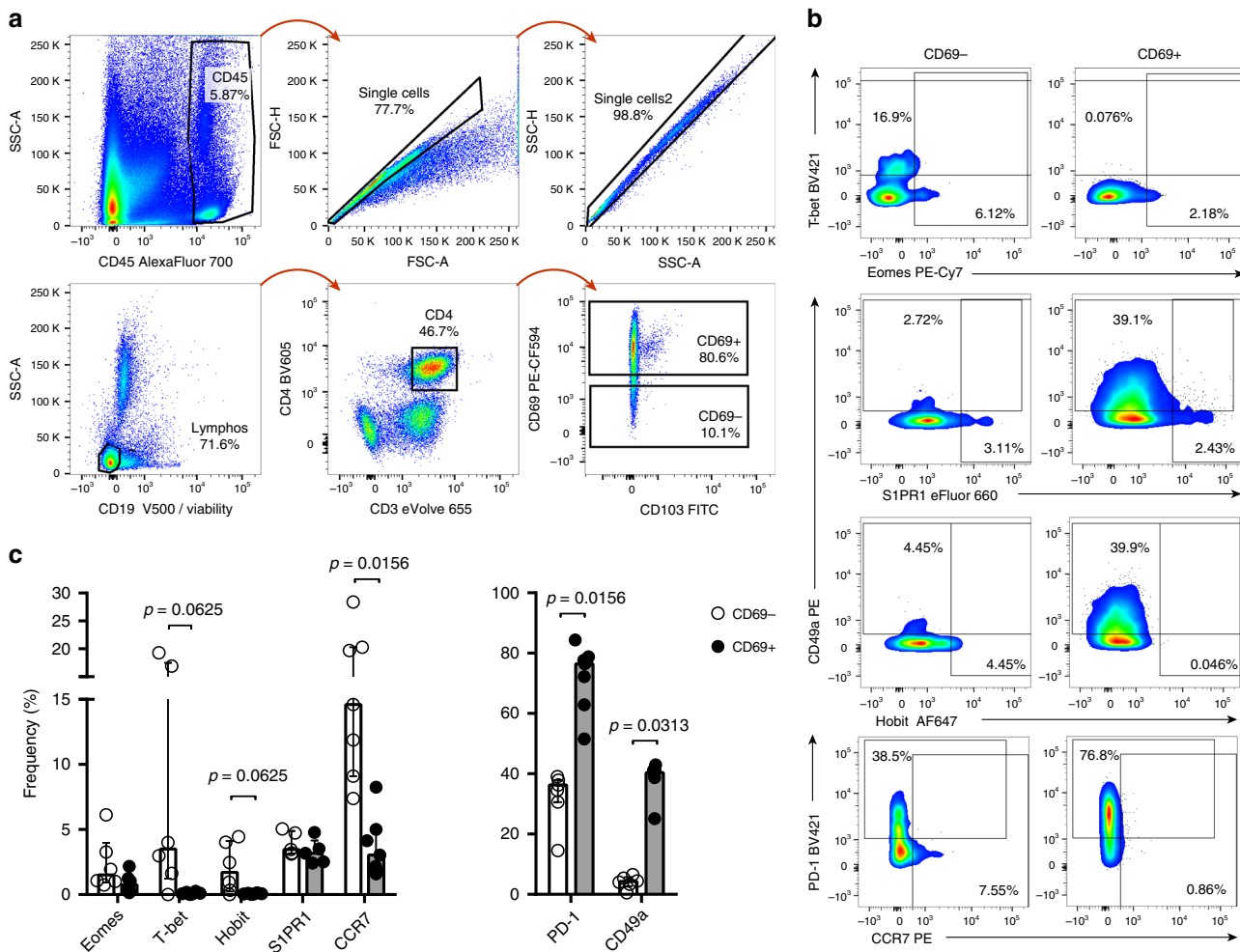

**Fig. 1** CD4$^+$ T$_{RM}$ identification in cervix. **a** General gating strategy for phenotyping of CD4$^+$ T cells obtained from cervicovaginal tissue of healthy donors. Gating strategy consisted of selecting hematopoietic CD45$^+$ cells, followed by a double doublet exclusion, dead and CD19$^+$ cells exclusion and finally a CD3$^+$ CD4$^+$ T cell gate from where CD69$^{+/-}$ cells were identified. **b** Representative flow cytometry plots of the expression of different cell-surface proteins and transcriptional factors in the CD4$^+$CD69$^{+/-}$ T cell subsets from the cervical tissue of healthy donors (CD69$^-$ on the left column, CD69$^+$ on the right column). **c** Frequency of different cell-surface proteins and transcriptional factors shown in **b** for CD4$^+$CD69$^-$ T cells (empty circles) and CD4$^+$CD69$^+$ T cells (full circles; $n = 5$ for S1PR1; $n = 6$ for Eomes, T-bet, Hobit, CD49a; $n = 7$ for CCR7, PD-1). Lines and error bars represent median and interquartile ranges. Statistical comparisons were performed using Wilcoxon matched-pairs signed rank test to compare the two groups. Source data are provided as a Source Data file

remarkably higher in CD69$^+$ CD4$^+$ T cells with values of around 40.4% for CD49a and of 76.45% for PD-1 compared to the CD69$^-$ fraction (median values of 4.26% for CD49a and of 36.3% for PD-1, $p = 0.0156$ for PD-1 and $p = 0.0313$ for CD49a). All together, these analyses confirmed that >95% of the CD4$^+$ T cells expressing CD69 in cervix are phenotypically associated to *bona fide* T$_{RM}$[25,31,32].

**CCR5$^+$ CD4$^+$ T$_{RM}$ protein expression signature in cervix.** In order to determine if CD4$^+$ T cells expressing CD69, from now on called T$_{RM}$, represent a reservoir for HIV, we first further characterized the expression profile of T$_{RM}$ as potential HIV-1 targets. Of note, for these analyses, we considered endocervix and ectocervix separately, since differences exist between these mucosal tissues in terms of structure, cellular components and ex vivo HIV replicative capacity[23]. Considering that HIV-1 isolates from newly infected individuals predominantly use CCR5 as a co-receptor, we performed a detailed analysis of the phenotype of cervical CD4$^+$ CCR5$^+$ CD69$^+$ (T$_{RM}$) and CD69$^-$ (non-T$_{RM}$) cells from HIV uninfected tissues. Moreover, since we observed

that a fraction of cervical CD4$^+$ T$_{RM}$ consistently expressed low levels of CD32 in uninfected tissues, a molecule which expression has been associated with the identification of transcriptionally active infected cells during HIV infection[10,11,13], we analyzed this fraction separately to determine phenotypic characteristics that could render these cells more vulnerable to HIV infection. Figure 2a shows the general gating strategy established for these analyses (which follows the gating already established in Fig. 1a). Since several recent papers focused on demonstrating that CD32 expression does not enrich for HIV-latently infected cells raised concerns on the purity of these cells[33–35] and, in order to eliminate potential contamination by B cells doublets (CD32 high subset), we only analyzed the fraction of CD4$^+$ T cells with dim expression of CD32. We used the levels of CD32 expression in B cells (indicated with a dotted black line, Fig. 2a) and of the corresponding fluorescence minus one control (Supplementary Fig. 1a) to define a narrow dim CD32$^+$ gate. Further, the existence of these cells in tissue was confirmed by Amnis imaging flow cytometry, which allowed visualization of individual CD4$^+$ CD69$^+$ (T$_{RM}$) cells expressing dim levels of CD32 (Supplementary Fig. 1b and 1c). These data together with recent publications

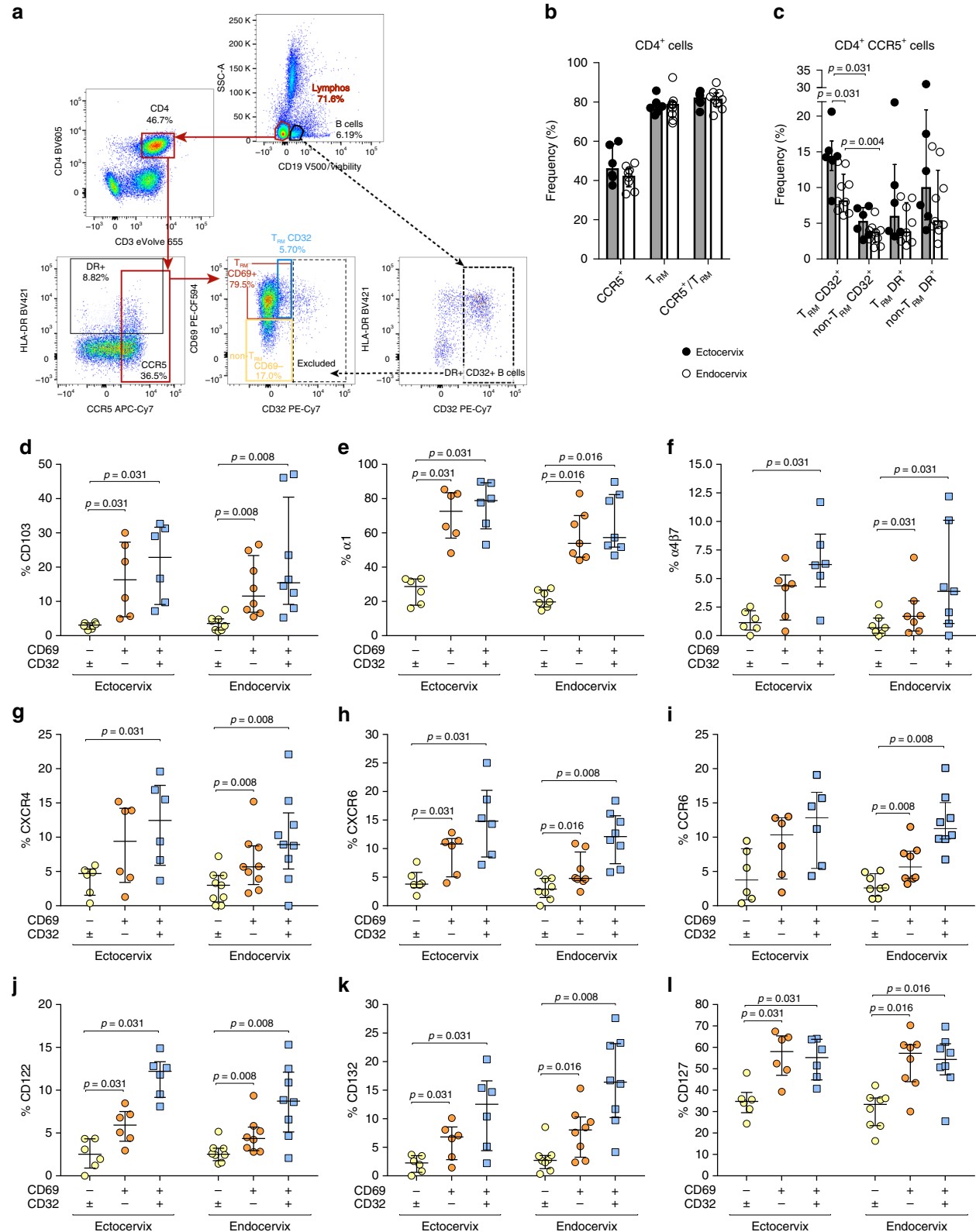

demonstrating that CD32–RNA correlates with HIV transcriptional activity in T cells from the gut[13], and from lymph nodes[10], not only confirms their existence in human tissues, but also raises interest on defining their nature.

Out of total CD4[+] T cells, a median of 45.8% expressed CCR5 in ectocervix and a median of 41.9% expressed CCR5 in endocervix (Fig. 2b). Further, the median frequency of $T_{RM}$

within total CD4[+] T cells or within CD4[+] CCR5[+] T cells was around 80% in both tissues (Fig. 2b). Also, a small subset of CD4[+] CCR5[+] $T_{RM}$ expressed HLA-DR or CD32, which in the case of CD32[+] were significantly more frequent in ectocervix than in endocervix ($p = 0.031$, Fig. 2c). Importantly, while HLA-DR expression was similar in CD4[+] CCR5[+] regardless of CD69 expression, with a median of 5.9% and 3.7% in $T_{RM}$ and of 9.9%

**Fig. 2** $CCR5^+$ $CD4^+$ $T_{RM}$ protein expression signature in cervix. **a** Gating strategy for phenotyping of $CD4^+$ T cells obtained from cervicovaginal tissue of healthy donors. Based on the gating strategy shown in Fig. 1a, a lymphocyte gate (marked in red) was selected in a plot where dead and $CD19^+$ were excluded. From one hand, $CD3^+$ $CD4^+$ T cells and then CCR5 positive cells were selected to further analyze three different subsets based on CD69 and CD32 expression: non-$T_{RM}$ (yellow gate), $T_{RM}$ $CD32^-$ (orange gate) and $T_{RM}$ $CD32^+$ (blue gate). Of note, to identify $CD4^+$ T cells expressing low levels of CD32 ($CD32^{dim}$, blue gate), we previously defined a gate of exclusion in putative B cells ($CD19^+$, depicted in dotted black lines) as these cells express high levels of CD32. **b** Frequency of diverse subsets within $CD4^+$ T cells from human ectocervix ($n = 6$) and endocervix ($n = 9$). **c** Frequency of diverse subsets within $CD4^+$ $CCR5^+$ T cells from human ectocervix ($n = 6$) and endocervix ($n = 9$). **d–l** Frequency of expression of diverse cell-surface proteins within three different $CD4^+CCR5^+$ T cell subsets; namely $CD69^-$ non-$T_{RM}$ (yellow), $CD32^-$ $T_{RM}$ (orange) and $CD32^+$ $T_{RM}$ (blue), in human ectocervix ($n = 6$) and endocervix ($n = 7$ for 2e, 2f; $n = 8$ for 2d, 2h, 2i, 2j, 2k, 2l; $n = 9$ for 2b, 2c, 2g). Lines and error bars represent median and interquartile ranges. All statistical comparisons were performed using Wilcoxon matched-pairs signed rank test. Source data are provided as a Source Data file

and 5.3% in non-$T_{RM}$ for ectocervix and endocervix respectively; CD32 expression was strongly associated to $T_{RM}$ compared to non-$T_{RM}$ (median of 14.3% and 8.1% in $T_{RM}$ *vs.* 5.2% and 3.7% in non-$T_{RM}$ for ectocervix and endocervix respectively; $p = 0.031$ for ectocervix and $p = 0.004$ for endocervix, Fig. 2c).

Then, we analyzed the expression of several surface markers related to HIV-susceptibility and persistence in a total of three different $CD4^+$ $CCR5^+$ T cell subsets: non-$T_{RM}$, $CD32^-$ $T_{RM}$ and $CD32^+$ $T_{RM}$ (Fig. 2a). Examples of individual markers are shown in Supplementary Fig. 2a. We observed that $T_{RM}$, including $CD32^+$ $T_{RM}$, expressed significantly higher levels of CD103 (integrin αE) than non-$T_{RM}$ in ectocervix ($p = 0.031$ in both cases) and mainly in endocervix ($p = 0.008$ in both cases, Fig. 2d). This was expected since CD103 is associated to intraepithelial retention of $CD8^+$ $T_{RM}$ and, to less extent, of $CD4^+$ $T_{RM}$[25,26]. Other integrins have also been associated with retention of $T_{RM}$ in tissues, and we observed that in cervical tissue, integrin α1 (CD49a) is highly associated to $CD4^+$ $CCR5^+$ T cells displaying a $T_{RM}$ phenotype ($p < 0.035$ for all comparisons, Fig. 2e). Additionally, we detected an enrichment of α4β7 expression on $CD32^-$ and $CD32^+$ $T_{RM}$ in the endocervix and on $CD32^+$ $T_{RM}$ in the ectocervix ($p = 0.031$ for all comparisons, Fig. 2f), indicating that some $T_{RM}$ could represent preferential targets for HIV-1[28,36]. Other HIV co-receptors, such as CXCR4 and CXCR6[37], were also more frequent in the $T_{RM}$ fractions, where $CD32^+$ $T_{RM}$ presented even higher frequencies compared to $T_{RM}$ not expressing CD32 (Fig. 2g, h). Expression of CCR6, a chemokine receptor associated to T helper 17 (Th17) cells, was also strongly associated to $T_{RM}$ and, mostly, to $CD32^+$ $T_{RM}$ from the endocervix, which again showed even higher expression compared to $CD32^-$ $T_{RM}$ ($p = 0.008$, Fig. 2i).

We also studied the presence of several interleukin receptors (IL), namely CD122, CD132 and CD127, as they may confer characteristics of longevity and homeostatic proliferation, with strong implication for maintenance of HIV reservoirs[5]. CD122 (β-chain) and CD132 (γ-chain) together form the IL-2 and IL-15 receptors, while CD127 (α-chain) and CD132 form the IL-7 receptor. We detected higher frequency of all three IL receptors in $CD4^+$ $CCR5^+$ $T_{RM}$ compared to non-$T_{RM}$ in both tissues ($p < 0.032$ for all comparisons, Fig. 2j–l). However, CD122 and CD132 expression was associated to a small fraction of $CD4^+$ $CCR5^+$ $T_{RM}$, which was higher in the $CD32^+$ $T_{RM}$ fraction compared to the $CD32^-$ fraction (Fig. 2j, k). In contrast, CD127 was expressed in a median of 34.8% of non-$T_{RM}$ and in > 55% of $CD32^{+/-}$ $T_{RM}$, confirming that, as occurs with other tissues[25], there is also retention of this receptor in memory cells from the cervical mucosa (Fig. 2l). Surprisingly, while the frequency of each IL-chain individually was remarkably higher in $T_{RM}$ compared to non-$T_{RM}$, there were few cells co-expressing CD132 and CD122, or CD127 and CD132 within the $CD32^-$ $T_{RM}$ fraction (Supplementary Fig. 2b and 2c). In contrast, a small but significant fraction of cells co-expressing these receptors was found in non-$T_{RM}$ compared to $T_{RM}$, while $CD32^+$ $T_{RM}$ showed even higher frequencies (Supplementary Fig. 2b and 2c). Further,

we also analyzed CCR7, which was minimally expressed in all subsets (<5% in general) and, as expected, more associated to non-$T_{RM}$ that will recirculate ($p = 0.031$, Supplementary Fig. 2d). CXCR3 expression was frequent in $CD32^-$ $T_{RM}$ and even more so in $CD32^+$ $T_{RM}$ of both tissues ($p < 0.032$ in all cases); CD161 was highly expressed in both $CD4^+$ $CCR5^+$ $T_{RM}$ cell fractions ($p < 0.032$ for each comparison), as shown before for other $T_{RM}$ phenotypes in tissue[26]; while CCR2 and γδ T cells were hardly detected in any of the subsets studied, except for the $CD32^+$ $T_{RM}$ fraction resident in the endocervix, which showed higher expression of CCR2 compared to non-$T_{RM}$ ($p = 0.031$, Supplementary Fig. 2e-h).

Together, these data not only confirm expression of some of the markers associated to the $T_{RM}$ phenotype in other tissues[25,26] in cervical $CCR5^+$ $T_{RM}$, but also strongly suggest two concepts related to our hypothesis: (1) $T_{RM}$ express several molecules that may render them highly susceptible to HIV infection (namely α4β7, CXCR4, CXCR6 and CCR6), and (2) these cells are potentially self-renewed and maintained long-term in tissues (i.e., CD122, CD132, CD127). Both concepts apply to $T_{RM}$ in general and, in particular, to a small fraction of $T_{RM}$ co-expressing low levels of CD32, which is even more enriched in most of these surface proteins. Thus, although CD32 expression in blood is associated to concomitant expression of CD69 indicating early activation[10,11], in cervical tissue $CD4^+$ T cells expressing CD32 are enriched in markers associated to truthfully $T_{RM}$ phenotypes. Overall, considering that the $T_{RM}$ phenotype is the predominant $CD4^+$ T cell subset in cervical mucosal tissues and their quiescent state, exemplified by low HLA-DR here or, as reported before, low Ki67[25], these cells are strong candidates to embody viral reservoirs in different peripheral tissues.

**$CD4^+$ $T_{RM}$ are preferentially infected during ex vivo infection.** In order to demonstrate that $CD4^+$ $T_{RM}$ from cervix represent a significant cellular reservoir, we first aimed to address if this subset supports infection ex vivo and which are the dynamics of CD69 after HIV infection in these tissues. For this, cervicovaginal tissues from healthy donors were infected with 7200 $TCID_{50}$ of an HIV-1$_{BaL}$ viral stock[38] for 10–12 days, and infection was measured by flow cytometry analysis of intracellular p24 staining and by cell-associated HIV-1 DNA in purified subsets. Since CD4 and CCR5 molecules are down-regulated by the accessory protein Nef during HIV-1 infection[39], we analyzed p24 viral antigen expression in a gate including $CD4^+$ and $CD4^-$($CD8^-$) T cells, as shown in the gating strategy (Fig. 3a). For each individual cervical explant infected, the frequency of p24$^+$ cells ranged from 1.92 to 13.4% of $CD4^{+/-}$ T cells (median of 7.3%), in all cases evidencing CD4 down-regulation (Fig. 3b). When analyzed by subsets, expression of p24 viral antigen was significantly higher in $T_{RM}$, $CD103^+$ $T_{RM}$ and $CD32^+$ $T_{RM}$ compared to non-$T_{RM}$ ($p = 0.004$, $p = 0.020$ and $p = 0.008$ respectively; Fig. 3b). This increase corresponded to a median of 5.02 times more infection in $T_{RM}$ compared to non-$T_{RM}$, and of 7.09 times more in the case of

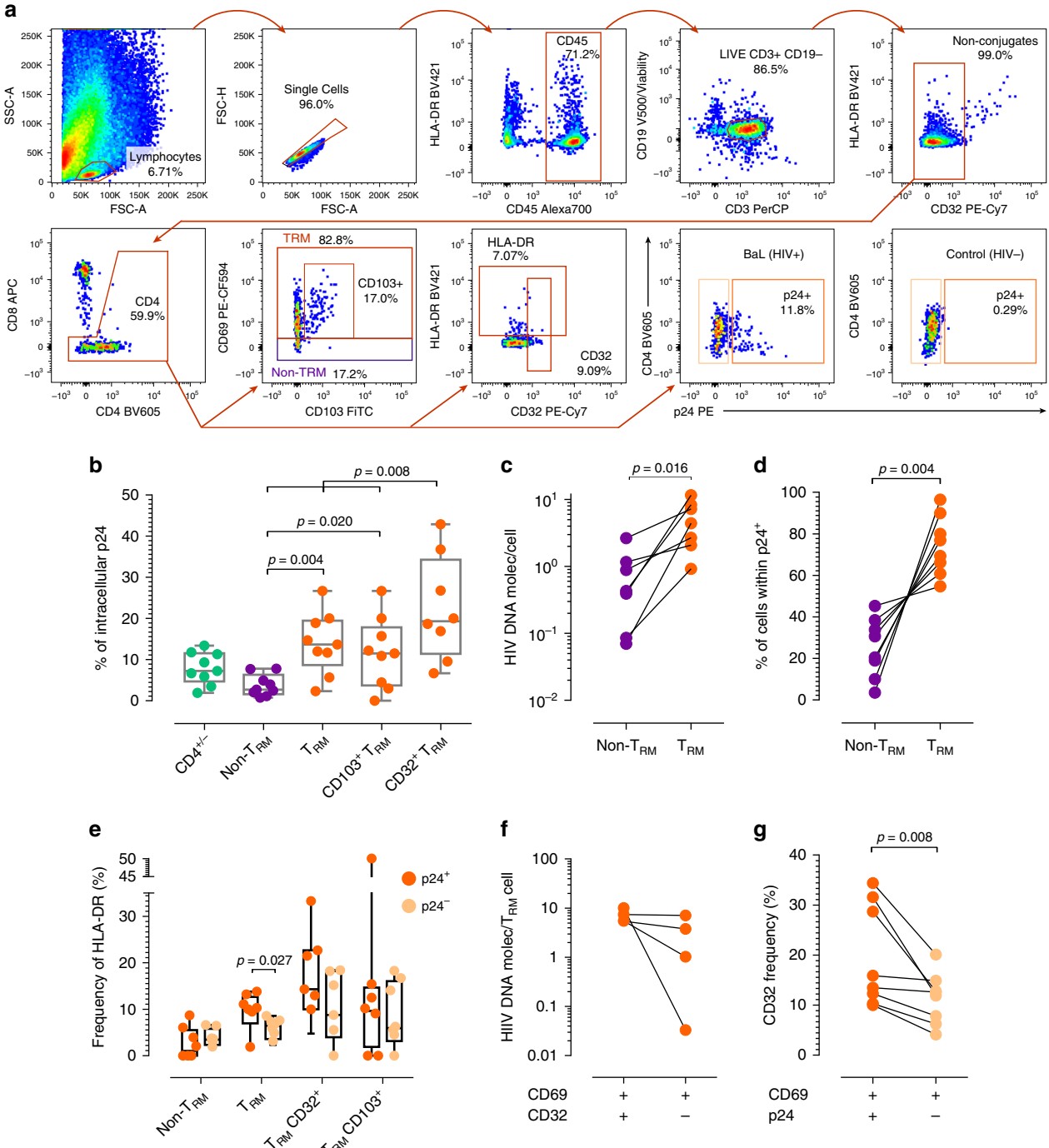

**Fig. 3** $T_{RM}$ are preferentially infected in the cervical explant model of HIV infection. **a** General gating strategy for phenotype analyses and intracellular p24 detection in CD4$^{+/-}$ $T_{RM}$ and non-$T_{RM}$ obtained from cervicovaginal tissue infected ex vivo with HIV-1$_{BaL}$ strain. Gating strategy consisted of a lymphocyte gate based on FSC vs. SSC, followed by doublet exclusion, hematopoietic CD45$^+$ cells, live CD3$^+$ T cells with CD19$^+$ cells exclusion, further B cell exclusion and a CD4$^{+/-}$ T cell gate from where the different gates shown were quantified: CD4$^{+/-}$ non-$T_{RM}$ (purple) or CD4$^{+/-}$ $T_{RM}$ (orange) or HLA-DR/ CD32$^{dim}$ expression. An example of p24 expression from an infected and a non-infected sample is shown. **b** Frequency of p24 viral antigen in total CD4$^{+/-}$ cells (green), CD4$^{+/-}$ non-$T_{RM}$ (purple), CD4$^{+/-}$ $T_{RM}$ (orange), CD4$^{+/-}$ CD103$^+$ $T_{RM}$ and CD4$^{+/-}$ CD32$^+$ $T_{RM}$ detected after 10–12 days of ex vivo infection and measured by flow cytometry ($n = 9$, except for CD4$^{+/-}$ CD32$^+$ $T_{RM}$ where $n = 8$). **c** Number of HIV-DNA molecules per cell in non-$T_{RM}$ and $T_{RM}$ detected after 10–12 days of ex vivo infection and measured by qPCR ($n = 7$). **d** Frequency of non-$T_{RM}$ and $T_{RM}$ from the total p24 positive cells measured by flow cytometry ($n = 9$). **e** Frequency of HLA-DR expression in different subsets based on their expression of p24 viral antigen after 10–12 days of ex vivo infection ($n = 9$, except for CD4$^{+/-}$ CD32$^+$ $T_{RM}$ where $n = 8$). **f** Number of HIV-DNA molecules per cell in $T_{RM}$ CD32$^+$ and $T_{RM}$ CD32$^-$ detected after 10–12 days of ex vivo infection and measured by qPCR ($n = 4$). **g** CD32 frequency in p24$^+$ $T_{RM}$ compared to p24$^-$ $T_{RM}$, measured by flow cytometry ($n = 8$). For **b** and **e** the median is shown as a solid line, the box indicates the 25–75 percentile range, while the whiskers show the range. Statistical analysis consisted of a Wilcoxon matched-pairs signed rank test. Source data are provided as a Source Data file

$CD32^+$ $T_{RM}$. Moreover, regardless of the frequency of infection, cell-associated HIV-1 DNA measured by qPCR was more abundant in $CD4^{+/-}$ $T_{RM}$ with a median of 4.4 vDNA molecules/cell compared to 0.43 in non-$T_{RM}$ ($p = 0.016$, Fig. 3c). This finding was also observed for productively infected cells, namely $CD4^{+/-}$ $p24^+$ cells, which contained higher proportion of $T_{RM}$ (>50% in all cases) than of non-$T_{RM}$ ($p = 0.004$, Fig. 3d).

Unexpectedly, HLA-DR expression in all subsets from these tissues remained very low, regardless of productive infection (Fig. 3e). As shown, the percentages of this marker were most often <20% in all subsets. Productive infection ($p24^+$) of total $T_{RM}$ was associated with a significant increase of HLA-DR expression compared to their uninfected counterparts ($p = 0.027$; Fig. 3e). Yet, these values were disproportionally lower than the levels observed on HIV infected cells from blood where, for example, $CD32^+$ $CD4^+$ infected cells have been shown to have frequencies of HLA-DR expression >80%[10]. These data suggest that in cervical tissue activation markers and their dynamics may not be comparable to blood. In fact, since HIV infection *per se* could induce up-regulation of CD69 on infected cells from peripheral blood, we determined the dynamics of CD69 expression and HLA-DR over 10 days of infection in cervical tissue (Supplementary Fig. 3a). Surprisingly, the frequency of CD69 expression decreased over time, with no significant changes in HLA-DR expression, similarly to what we observed in the concomitant non-infected control.

In addition, we separated $CD69^+$ and $CD69^-$ $CD4^+$ T cells from fresh cervical suspensions, which we immediately infected to evaluate infection (p24) and CD69 expression. From a total of four individual tissues, a median of 3.23% $CD69^+$ were $p24^+$ 3 days after infection, while only in one out of four tissues we detected few p24 positive cells (0.21%) in the $CD69^-$ fraction (Supplementary Fig. 3b). Moreover, in these experiments we detected minimal enhancement of CD69 expression in the $CD69^-$ fraction (Supplementary Fig. 3b). To further confirm the residency nature of most of the cervical cells supporting infection, we stimulated 10 day-infected tissue blocks with CCL19, CCL21, and S1P overnight to attract non-$T_{RM}$ out of the tissue in a transwell migration assay. CCL19 and CCL21 are chemokine-ligands attracting CCR7 expressing cells, while S1P promote egress of cells expressing $S1PR1$[40]. Next day, we determined the level of infection in tissue blocks, as well as in the supernatant (Supplementary Fig. 3c). This experiment demonstrated higher frequency of $p24^+$ cells retained within the tissue compared to the supernatant (Supplementary Fig. 3c). In addition, CD69 expression in total $CD8^-$ T cells was higher within the tissue (~60–81%) than in the supernatant (~35–52%). Interestingly, while productive infection was again strongly associated to the $T_{RM}$ phenotype in the tissue (with >72% of the $p24^+$ cells expressing CD69), most of these infected cells did not express the α-chain of the IL-7 receptor, CD127, also associated to the $T_{RM}$ phenotype in healthy cervical tissue (Supplementary Fig. 3c).

Lastly, in four of these cervicovaginal explants infected ex vivo, in which, after tissue processing, a high number of T cells were obtained, we further purified $CD4^{+/-}$ $T_{RM}$ expressing CD32 to determine their vDNA content. Although limited by the small number of experiments, there was a trend towards higher content of vDNA per cell in $CD32^+$ *vs.* $CD32^-$ $CD4^{+/-}$ $T_{RM}$ (Fig. 3f). In addition, although most of the $p24^+$ cells were not $CD32^+$, CD32 expression was significantly more frequent in $p24^+$ $T_{RM}$ compared to $p24^-$ $T_{RM}$ ($p = 0.008$, Fig. 3g). Taken together, these results indicate that $CD4^+$ $T_{RM}$ preferentially support HIV infection ex vivo, as demonstrated by higher vDNA content per cell and by high expression of intracellular p24 viral antigen. These data concur with recently reported high CD69 expression on infected tonsillar T cells and predicted precursor cells,

indicating preferential infection of these cells, as opposed to HIV-induced changes in receptor expression[41]. It remains to be elucidated if CD32 expression on $CD4^+$ $T_{RM}$ remains stable in the HIV-1 infected cells over the virus life cycle or is modulated by infection, and mostly associated to a fraction of activated cells as proposed[10].

**HIV infection impacts on cervical $CD4^+$ $T_{RM}$ in women.** Next we determined if $CD4^+$ $T_{RM}$ remain in tissue from HIV infected patients, since they may be impaired as a consequence of the retroviral infection, as suggested[42]. Thus we compared cervical $CD4^+$ T cells from two groups of patients: cervicovaginal tissue of previously analyzed normal healthy donors (ND) and ART-suppressed $HIV^+$ patients (Table 1). As expected, the frequency of $CD4^+$ cells out of total $CD45^+$ lymphoid-cells decreased from a median of 32.2% in ND to about 20.2% in $HIV^+$ patients ($p = 0.002$, Fig. 4a). Importantly, the frequency of $T_{RM}$ within the same fraction was also lower in $HIV^+$ patients compared to ND ($p = 0.006$, Fig. 4a). Further, the median of $CD32^+$ $T_{RM}$ out of the $CD45^+$ lymphoid-cells was of 0.52% in $HIV^+$ patients compared to 1.87% in ND ($p = 0.006$, Fig. 4a). Additionally, cervical $CD4^+$ $HLA-DR^+$ T cells were proportionally increased in $HIV^+$ women with a median of 3.15% compared to 1.29% in ND (Fig. 4a), similar to what has been described[20].

When we determined the frequency of HLA-DR in different fractions, we observed that in virally suppressed $HIV^+$ patients, HLA-DR expression was significantly enhanced compared to ND in total $CD4^+$ $T_{RM}$ ($p < 0.0001$), $T_{RM}$ $CD103^+$ ($p = 0.01$), and in non-$T_{RM}$ ($p = 0.001$), suggesting a higher degree of activation of all these cells in infected compared to uninfected tissues (Fig. 4b). In contrast, the frequency of CD32 in the different subsets was similar between the two groups, except for a trend to lower frequencies of $CD32^+$ $CD103^+$ $T_{RM}$ cells in virally suppressed $HIV^+$ patients compared to ND ($p = 0.0756$; Fig. 4c). However, as mentioned earlier, it was clear that CD32 was strongly associated to the $CD4^+$ $T_{RM}$ phenotype, since higher percentages of CD32 expression were detected in total $CD4^+$ $T_{RM}$ and $CD4^+$ $CD103^+$ $T_{RM}$ of both groups compared to their corresponding non-$T_{RM}$ fractions ($p < 0.05$ for all comparisons, Fig. 4c). Together, these data suggest a direct impact of HIV infection on the $CD4^+$ $T_{RM}$ fraction of the lower genital tract, as evidenced by its depletion and activation, which remains after >10 years of effective ART suppression in $HIV^+$ women.

**$T_{RM}$ are the main cellular reservoir in the cervical mucosa.** Tissue body reservoirs of HIV-1 or SIV have mainly focused on the analyses of virus burden in blood, lymphoid tissues and gastrointestinal tract[3,16], while other mucosal tissues such as the female genital tract have not been accurately evaluated for HIV-1 persistence. To address if the cervical mucosa is a site of viral persistence and if mucosal $T_{RM}$ represent a cellular reservoir, we compared contemporaneous blood and cervical samples from 8 HIV-infected women who had been ART-suppressed for at least a year, with a median of 4 years of suppression (Table 1). First, we compared the total amount of vDNA in total $CD4^+$ and $CD4^-$ ($CD8^-$) T cells from tissue and blood, using the same gating strategy shown for the ex vivo model to include infected $CD4^+$ T cells that down-regulate this receptor (Fig. 3a). Importantly, cervical tissue contained significantly more HIV-DNA copies per $10^6$ $CD4^{+/-}$ T cell than blood ($p = 0.015$; Fig. 5a). While $CD4^{+/-}$ T cells from blood harbored between 28 and 4812 vDNA copies/ $10^6$ $CD4^{+/-}$ T cells, cervical cells contained between 1366 and 23,270 copies per million cells, which corresponded to fold changes of up to >200 times more provirus per cell in cervix than in blood (Fig. 5a). Of note, the number of sorted cells highly

**Table 1 Clinical data of HIV-infected patients included in the study**

| # Patient ID | Analyses | Age (yr) | Time since HIV diagnosis (yr) | Viral load (copies/ml) | CD4 cell count (cells/μl) | Time on ART-suppressed (yr) | ART regimen |
|---|---|---|---|---|---|---|---|
| M00 | DNA | 55 | >12 | <50 | 566 | >10 | TDF + FTC + ETR |
| M01 | Flow | 51 | 3.41 | <50 | 420 | 3 | 3TC + ABC + DTG |
| M02 | DNA/Flow | 52 | 27 | <50 | 940 | 13.58 | 3TC + ABC + RAL |
| M03 | DNA/Flow | 54 | 16.58 | <50 | 1260 | 16 | TAF + FTC + EVG/c |
| M04 | DNA/Flow | 26 | 4 | <50 | 520 | 3.67 | 3TC + ABC + DTG |
| M05 | DNA/Flow | 57 | 27 | <50 | 1400 | 15.33 | TDF + FTC + EVF |
| M11 | Flow | 44 | 13.16 | <50 | 440 | 11.66 | 3TC + DRV/c |
| M15 | Flow | 48 | 29.50 | <50 | 380 | 16.50 | 3TC + ABC + DTG |
| M16 | Flow | 49 | 30.50 | <50 | 490 | 2.64 | DRV/c + DTG |
| M17 | Flow | 47 | 26.58 | <50 | 820 | 20.67 | 3TC + ABC + DTG |
| M18 | Flow | 63 | 19.54 | <50 | 160 | 18.45 | 3TC + DTG |
| M19[a] | Flow | 51 | 31.50 | <50 | 200 | 6.82 | DRV/c + DTG |
| M20[a] | DNA/Flow | 53 | 29 | <50 | 1257 | 1.46 | ATV/r + 3TC + ddI |
| M21 | Flow | 49 | 22.79 | <50 | 850 | 14.66 | 3TC + ABC + NVP |
| M22 | DNA/Flow | 48 | 26.50 | <50 | 850 | 2.55 | 3TC + ABC + DTG |
| M23[a] | DNA/Flow | 46 | 23.46 | <50 | 2850 | 2.09 | 3TC + DTG |
| M25 | DNA/Flow | 48 | 17.62 | <50 | 1040 | 13.87 | EVG/c + FTC + TAF |
| M26 | Flow | 44 | 26 | <50 | 511 | >15 | 3TC + ABC + DTG |
| M27[a] | Flow | 54 | 34.39 | <50 | 1380 | 0.71 | DRV/c + RPV |
| M28 | DNA | 54 | 20 | <50 | 750 | 4.5 | 3TC + ABC + DTG |
| P1 | RNA | 34 | 9.5 | 5100–10,400 | 760 | — | UNTREATED |
| P2 | RNA | 30 | 3.42 | <50 | 380 | 3.16 | 3TC + ABC + DRV + RTV |
| P3 | RNA | 49 | 4.25 | <50 | 910 | 4 | 3TC + ABC + DTG |
| P4 | RNA | 49 | 0.5 | <50 | 280–350 | 0.25 | TAF + FTC + EVG/c |
| P5[a] | RNA | 47 | 24.16 | <25 | 811 | 4 | EFV + FTC + TDF |
| P6 | RNA | 51 | 11.25 | <25 | 215 | 2 | EFV + FTC + TDF + EFV |
| P7 | RNA | 46 | 20.16 | <25 | 828 | 5.5 | DRV |

[a]All infections correspond to diagnosed heterosexual HIV-1 transmission except M19, M20, M23, M27, and P5, which used to be injecting drug-users. *TDF* tenofovir disoproxil fumarate, *FTC* emtricitabine, *ETR* etravirine, *3TC* lamivudine, *ABC* abacavir, *DTG* dolutegravir, *RAL* raltegravir, *TAF* tenofovir alafenamide fumarate, *EVG/c* elvitegravir boosted with cobicistat, *EVF* efavirenz, *NVP* nevirapine, *DRV/c* darunavir boosted with cobicistat, *DRV* darunavir, *RTV* ritonavir, *RPV* rilpivirine, *ATV* atazanavir, *ddI* didanosine

varied based on the sample size and the patient, and probably limited detection of vDNA in tissue from other patients (Supplementary Table 1). In fact, 15 more samples were also processed, but the low yield obtained after tissue processing limited performing any of these analyses. Nonetheless, patients with detectable vDNA in cervix had undetectable viral load in blood for at least 1 year and up to >11 years (Table 1). In cervical tissue of seven of these patients, by sorting $CD69^+$ $T_{RM}$ and $CD69^-$ non-$T_{RM}$, we demonstrated that the vDNA detected in these patients was associated to the $T_{RM}$ fraction in all but one patient, who had detectable vDNA in both $T_{RM}$ (5073 copies/$10^6$) and non-$T_{RM}$ cells (1606 copies/$10^6$). Of note, although in most tissues $T_{RM}$ were purified in higher numbers than non-$T_{RM}$, in patient #M04 higher numbers of $CD4^{+/-}$ non-$T_{RM}$ were sorted (Supplementary Table 1), yet only the $CD4^{+/-}$ $T_{RM}$ fraction was positive. Thus, we observed that the $T_{RM}$ fraction contained significantly more vDNA than the whole population of $CD4^{+/-}$ T cells present in cervical tissue ($p = 0.015$; Fig. 5b and Supplementary Fig. 4a). Next, we calculated the global contribution of $T_{RM}$ and non-$T_{RM}$ to the total pool of infected $CD4^{+/-}$ T cells in cervix of HIV-infected women. For the fractions that were undetectable for vDNA, we used the limit of detection of the assay based on the individual cellular inputs and the resulting number was plotted as the maximum possible contribution. We found that the $T_{RM}$ subpopulation significantly contributed to the pool of infected cells by representing a median of 95.65% compared to 4.34% of non-$T_{RM}$ ($p = 0.015$; Fig. 5c), which was partially due to their different frequencies (a median of 83.20% and 16.79%, respectively).

In contemporary blood samples from the same patients, we used the same flow cytometry panel to isolate four different fractions from total $CD4^{+/-}$ T cells: $CD69^+$ (in blood identifying recently activated cells), $CD103^+$ (a potential surrogate marker of tissue-derived T cells[43,44]), $CD32^+$ (here not CD69 or CD103) and $CD4^{+/-}$ T cells not expressing any of these three markers (Fig. 5d). We detected similar median levels of vDNA content in all subsets and no statistical differences were observed among them (Fig. 5d), regardless of different sorted cell numbers (Supplementary Table 1). However, when we analyzed the contribution of the different subsets to the blood reservoir, most of the contribution to the total pool of HIV in $CD4^{+/-}$ T cells was due to $CD69^-CD103^-$ $CD32^-$ $CD4^{+/-}$ T cells, since they were the most frequent subset, representing >96% of the $CD4^{+/-}$ T cells ($p = 0.0002$) (Fig. 5e). It is noteworthy that when we analyzed the phenotype of these PBMC from $HIV^+$ patients, we detected that in blood CD32 was also strongly associated to the CD69 fraction, in which a median of 38.2% of the cells expressed CD32 ($p = 0.004$ compared to total $CD4^{+/-}$ T cells, Supplementary Fig. 4b). On the other hand, HLA-DR protein expression was enriched in the $CD69^+$ and the $CD103^+$ fractions ($p = 0.008$ for $CD69^+$, $p = 0.004$ for $CD103^+$ compared to total $CD4^{+/-}$ T cells, Supplementary Fig. 4b). This implies that we partially decreased the total frequency of $CD32^+$ $CD4^{+/-}$ T cells by gating on $CD69^+$ $CD4^{+/-}$ T cells (which represented a median of 8.1% of the total CD32), thus underestimating their contribution to the total reservoir in blood. Simultaneously, the contribution of the CD69 fraction to the total pool of cells is strongly sustained by its high expression of CD32. Further

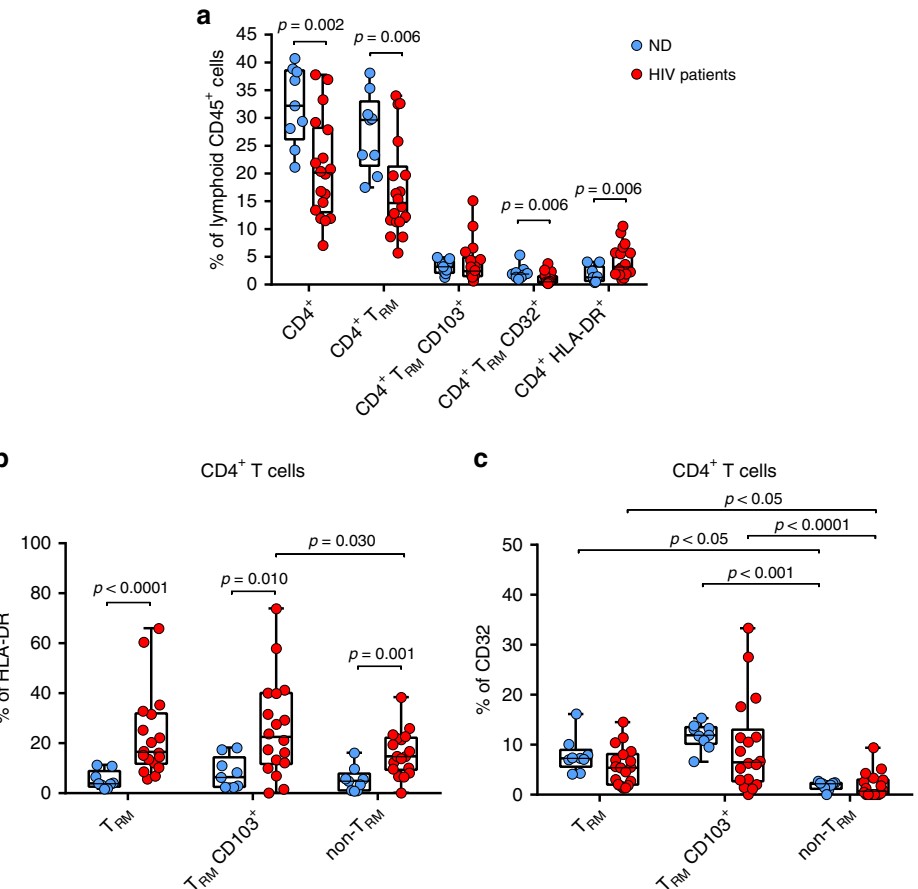

**Fig. 4** Impact of HIV infection on CD4+ $T_{RM}$ cell subsets from cervix. **a** Frequency of cervical CD4+, $T_{RM}$, $T_{RM}$ CD103+, $T_{RM}$ CD32+, and HLA-DR+ CD4+ T cells from the total cervical CD45+ lymphoid cells was determined following the gating strategy described in Fig. 3 in normal donors (ND, blue circles, n = 9) and HIV infected patients (red circles, n = 18, corresponding to M01-M05, M11, M15-M23, and M25-M27). **b** HLA-DR expression in cervical $T_{RM}$, CD103+ $T_{RM}$ and non-$T_{RM}$ in normal donors and HIV infected patients. **c** CD32 expression in cervical $T_{RM}$, CD103+ $T_{RM}$ and non-$T_{RM}$ in normal donors and HIV infected patients. The median is shown as a solid line, the box indicates the 25–75 percentile range, while the whiskers show the range. Statistical comparisons were performed using two-tailed Mann–Whitney rank test to compare the two groups and using Friedman test with post hoc Dunn's correction for multiple comparisons for intra-group comparisons. Source data are provided as a Source Data file

analyses discriminating these two fractions should determine their separate real contribution to the viral reservoir in blood.

Finally, by including CD4− (CD8−) T cells in these analyses we enriched the fraction of activated cells, since CD4− T cells from blood expressed significantly more CD69, HLA-DR, CD103 and CD32 than CD4+ T cells (p = 0.004 for CD69 and HLA-DR, p = 0.012 for CD103, p = 0.019 for CD32, Supplementary Fig. 4c), while in cervix only CD32 was associated to the CD4− fraction of T cells, yet HLA-DR had a strong opposite trend (p = 0.004 for both, Supplementary Fig. 4d). Of note, in terms of general expression, both CD32 and HLA-DR frequencies were higher in cervicovaginal cells than in blood of HIV+ patients, indicating that a fraction of tissue-resident cells are potentially more activated than circulating cells. In any case, altogether, these results provide evidence that CD4+ $T_{RM}$ are infected in vivo and represent an HIV target, at least in cervical tissue. Importantly, regardless of effective ART-suppression for more than 10 years, high levels of proviral DNA are still detected in CD4+ $T_{RM}$ from the cervical mucosa of HIV+ women, consistently suggesting that this cell type could represent an important cellular reservoir elsewhere.

**HIV−RNA+ CD69+ cells are detected in cervical tissue.** We used in situ hybridization (ISH) to determine if cervical tissue harbors vRNA+ cells, as performed before[10,45,46], and if these cells

co-express CD69. We performed this by combining ISH for HIV-1 viral RNA with immunohistochemistry of CD69, since CD69 mRNA expression was hardly detectable (data now shown). Cervical preparations from one untreated viremic controller and six ART-suppressed women were examined (Table 1, #P1-P7). In the cervical mucosa of the viremic controller patient we detected vRNA+ cells co-expressing CD69 in both the epithelium and the lamina propria (Fig. 6a). As shown in the associated pie charts, from the total of 15 HIV-vRNA+ cells detected in this sample, close to 73% were located in the submucosa and the rest in the epithelium (Fig. 6a). Although a lower proportion of CD4+ $T_{RM}$ are found in the intraepithelial compartment compared to CD8+ $T_{RM}$, CD4+ $T_{RM}$ are retained in both compartments[47]. Interestingly, some of the vRNA+ cells in the submucosa were associated to CD69 enriched zones (45% of them), which could concur with previously described semi-organized lymphoid structures containing resident memory lymphocyte clusters[48,49]. Of note, the strong staining of the HIV-1 RNA probe in some positive cells precluded distinctive determination of CD69 positivity and thus we did not count double positive cells in each compartment. Importantly, when we examined cervical preparations of ART-suppressed HIV+ women, we also detected vRNA+ cells expressing CD69 in the epithelium and the submucosa (Fig. 6b). In these patients the overall mean frequency of HIV-vRNA+ cells was of 0.114 cells/mm², which was similar to what was observed in the

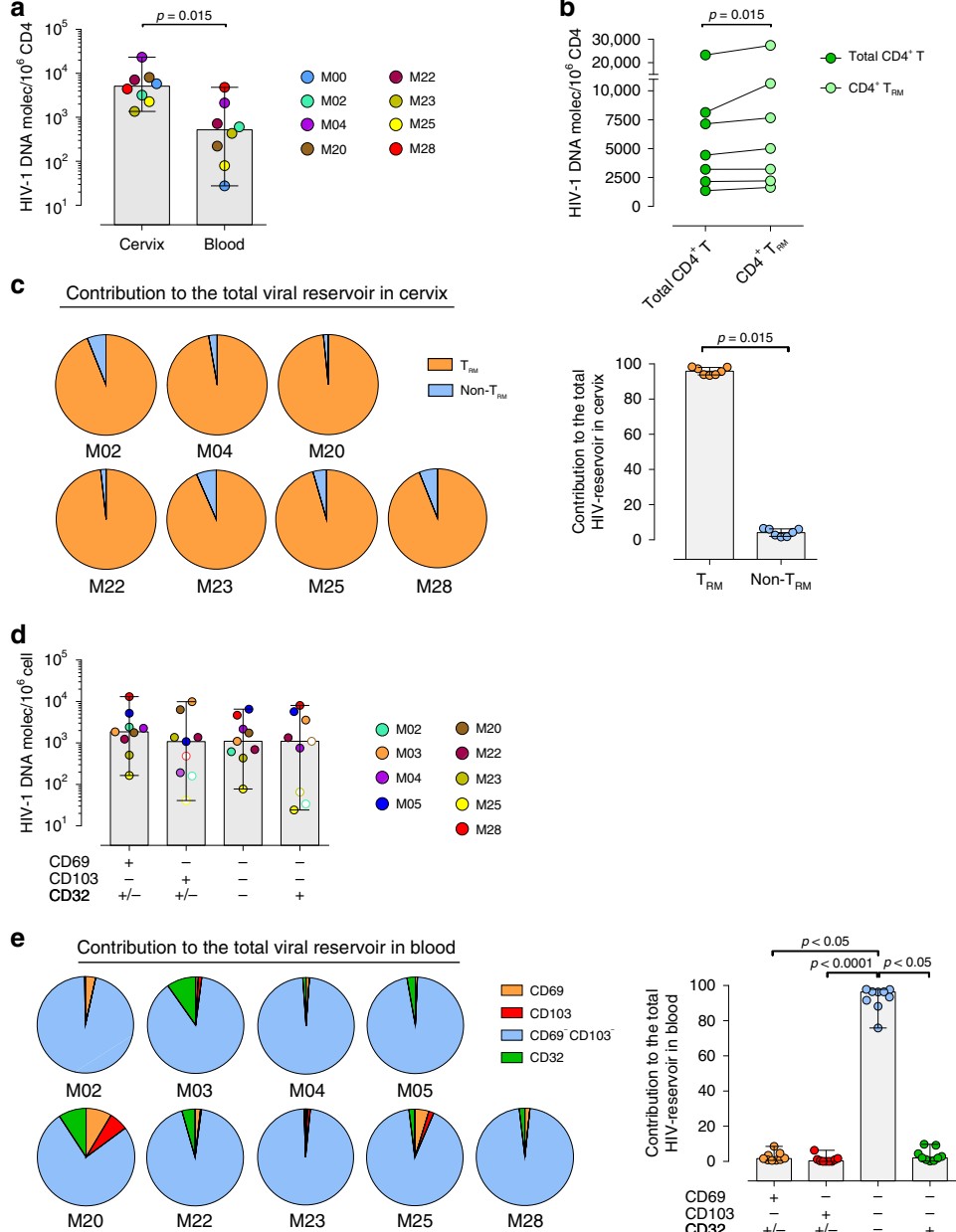

**Fig. 5** Cellular reservoirs in CD4$^{+/-}$ T cells from cervix and blood of ART-suppressed HIV-1 infected women. **a** Comparison of HIV-1 DNA molecules per million CD4$^{+/-}$ T cells in cervix and PBMCs from the same ART-treated women ($n = 8$). **b** Comparison of HIV-1 DNA molecules per million CD4$^{+/-}$ T cells and per million CD4$^{+/-}$ CD69$^+$ (T$_{RM}$) in cervix from seven ART-suppressed women. **c** Individual contribution of CD4$^{+/-}$ T$_{RM}$ and non-T$_{RM}$ to the total HIV reservoir in cervicovaginal tissue (left) and summary data from all patients (right). **d** HIV-1 DNA molecules per million cells in four different subsets based on the expression of CD69, CD103 and CD32 in CD4$^{+/-}$ T cells from blood in HIV-infected women ($n = 9$). Empty circles represent values under the limit of detection. **e** Contribution of analyzed subsets to the total HIV reservoir in blood (left) and summary data from all patients (right). Boxes and error bars represent median and range. Statistical comparisons in Fig. 5a–c were performed using Wilcoxon matched-pairs signed rank test and in Fig. 5d and e using Friedman test with post hoc Dunn's correction for multiple comparisons. Source data are provided as a Source Data file

cervical tissue from the viremic controller woman, where we detected 0.106 cells/mm². In five ART-suppressed HIV infected women where the epithelium was preserved, >70% of the vRNA$^+$ cells were found in the submucosa of the cervix, in general less frequently associated to high density CD69$^+$ staining, as shown in the pie charts for #P2 (Fig. 6b). Individual quantification and distribution of vRNA$^+$ cells in different tissue areas is shown in Fig. 6c, d for all patients. In any case, these analyses demonstrate substantial residual viral transcription in the cervical mucosa of ART-suppressed women often associated to CD69$^+$ and thus to T$_{RM}$.

## Discussion

Here we assert the existence of an overlooked HIV-1 reservoir in cervical tissue from women, since we detected substantial levels of HIV-1 DNA and RNA in the mucosa of ART-treated patients with undetectable plasma viral load for up to 10 years. Moreover, we identify CD4$^+$ T$_{RM}$ as a critical cellular reservoir that sustains persistence in peripheral tissues. This subset, recently characterized in multiple tissues, includes highly HIV-susceptible phenotypes in the female genital tract, which are simultaneously shaped for long-term permanence in tissues. Accordingly, and considering the proportion of CD4$^+$ T cells estimated to belong to

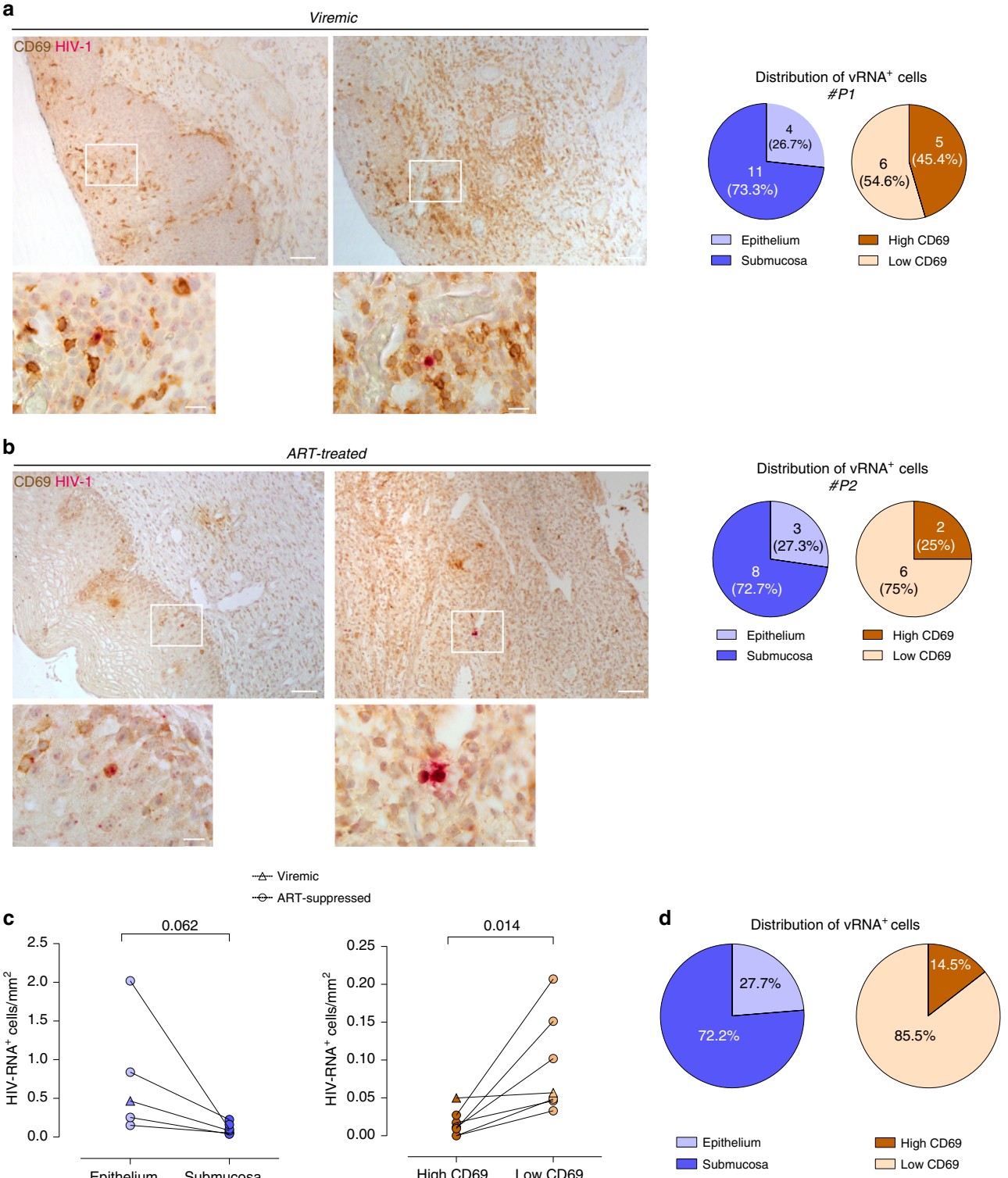

the $T_{RM}$ phenotype in tissues all over the human body[25,26], the contribution of this subset to HIV-1 persisting reservoirs may be of great importance. In fact, among these tissues with large proportions of $T_{RM}$ we find lymphoid and mucosal sanctuaries such as the gut, estimated to be the main reservoirs during treatment[3]. Thus, dissecting $CD4^+$ $T_{RM}$ heterogeneity in relation to HIV pathogenesis and determining the relative weight of specific $CD4^+$ $T_{RM}$ subsets as cellular reservoirs should be a priority for the HIV field.

$T_{RM}$ longevity in diverse sites may reflect the accumulation of antigenic experiences over the human lifespan[25], which may be particularly relevant in peripheral tissues that represent a mucosal barrier against a broad range of microbes, such as the female genital tract. However, $T_{RM}$ of the cervicovaginal mucosa of women have not been characterized in depth, regardless of being a site of exposure to many viral and bacterial infections with very high global burden. Recent efforts to define transcriptional and functional signatures of $CD69^+$ $T_{RM}$ have revealed consistent

**Fig. 6** Detection of HIV-RNA$^+$ CD69$^+$ cells in the cervical mucosa of virologically suppressed women. Paraffin-embedded cervical tissue from one viremic and six ART-suppressed HIV-infected women were stained for HIV-1 RNA (rose) using the RNAscope 2.5 HD Duplex Reagent Kit in combination with immunohistochemistry of CD69 protein (brown). **a** Two representative images of a viremic HIV-infected patient (#P1) with a cell co-expressing HIV-1 RNA and CD69 in the epithelium (left) and in the submucosa (right) are shown inside a white square. Lower panels show high magnification of each of these squares. Scale bar is 100 μm and 20 μm in top and bottom panels, respectively. Left pie chart shows the proportion of HIV-RNA$^+$ cells detected in the epithelium (light blue) or in the submucosa (dark blue) from the total of 15 HIV-RNA$^+$ cells detected in the cervix of #P1. Right pie chart shows the proportion of HIV-RNA$^+$ cells detected in the submucosa associated to CD69 enriched clusters (dark brown) or independent from CD69 clusters (light brown). **b** Two representative images of an ART-suppressed HIV-infected patient (#P2) with a cell co-expressing HIV-1 RNA and CD69 in the epithelium (left) and in the submucosa (right) are shown inside a white square. Lower panels show high magnification of each of these squares. Scale bar is 100 μm and 20 μm in top and bottom panels, respectively. Pie charts represent the same proportions as in **a** for a total of 11 HIV-RNA$^+$ cells detected in the cervix (left) or of 8 HIV-RNA$^+$ cells in the submucosa (right) of #P2 patient. **c** Comparison of positive cells harboring vRNA per square millimeter in the epithelium and the submucosa of $n = 7$ HIV-infected patients. Triangle symbols correspond to a viremic patient and circle symbols correspond to data from ART-suppressed patients. Statistical comparisons were performed using Wilcoxon matched-pairs signed rank test. **d** Distribution of HIV-RNA$^+$ cells in different localizations of the cervical tissue in $n = 7$ HIV-infected patients. Source data are provided as a Source Data file

patterns across human tissues, but also increased heterogeneity of the CD4$^+$ T$_{RM}$ fraction compared to CD8$^+$ T$_{RM}$[25,26]. Here we confirm that expression of CD69 in human cervical mucosa associates to a protein signature identifying *bona fide* T$_{RM}$ in >95% of these cells[25,26,31,32], which suggests residency associated to most CD4$^+$ T cells from the human female genital tract, as recently reported in the mouse model[30]. Importantly, we report that cervical CD4$^+$ T$_{RM}$ from both ectocervix and endocervix frequently express several surface proteins associated to increased HIV susceptibility[28,36,50,51]. In fact, CD4$^+$ T cells derived from the cervix of HIV$^-$ women expressing α4β7, α4β1, or the same CD69, were already shown to represent preferential targets of HIV entry[28]. Our study extends these findings, revealing that the majority of CCR5$^+$ CD4$^+$ T cells indeed belong to the T$_{RM}$ phenotype, which is highly enriched in the expression of α4β7 and α4β1 integrins, as suggested[50], but also in HIV-1 co-receptors such as CXCR4 or CXCR6[37]. Further, CCR6 expression, which may indicate a Th17 response, was also more frequent in CD4$^+$ CCR5$^+$ T$_{RM}$ from the endocervix. Importantly, Th17 cells not only represent preferential targets after SIV vaginal transmission[51], but also are enriched for replication-competent HIV-1 DNA in colon and blood from ART-treated patients[52].

Critically, we confirmed that T$_{RM}$ are indeed preferentially infected in human cervical tissue explants infected with HIV$_{BaL}$, where this phenotype harbored more vDNA and was more productively infected than their counterparts, CD69$^-$ non-T$_{RM}$. As mentioned, by using high-dimensional CyTOF analysis, Cavrois et al. recently demonstrated preferential infection of tonsillar CD69 positive cells, where predicted precursors of the HIV-infected cells had similarly high expression levels as the infected cells[41]. Although we cannot discard that non-T$_{RM}$ present in the cervical tissue may upregulate CD69 upon HIV infection, the preference to support HIV-infection of sorted CD69$^+$ cervical CD4$^+$ T cells, the lack of CD69 up-regulation after HIV-exposure in the sorted CD69$^-$ fraction, the dynamics of the expression of CD69 and HLA-DR during infection overtime, the infection level observed in CD103$^+$ T$_{RM}$, and the fact that most of the productive infection is retained within the tissue even after stimulation with S1P and CCR7-ligands, strongly suggest an important role for T$_{RM}$ supporting HIV infection, as implicit on the work from others[18,27,28]. The down-regulation of CD69 detected in cervical tissue culture has already been observed in the tonsil explant model[53] as well as in adipose tissue[18], and could be related to disruptions on cell trafficking and retaining stimulus. However, in the tonsil explant model, there was an increase in activation measured by HLA-DR and CD25 expression over time[53] that was not observed in our results. Moreover, although culturing tissue may affect retention signals, CD69 expression remained higher within the tissue blocks compared to the cells

that emigrated in the supernatant, confirming its association with mucosal retention. Importantly, after 10 days of infection, most of the productively infected cells remained associated to the T$_{RM}$ phenotype within the tissue. Thus, in principle, CD4$^+$ CCR5$^+$ T$_{RM}$ from a wide range of peripheral mucosal and lymphoid tissues may represent preferential targets of HIV infection.

Interestingly, we detected a small subset of cervical CD4$^+$ CCR5$^+$ T$_{RM}$ that readily expressed CD32, a marker originally associated to HIV-latently infected cells[9], which was later linked to transcriptionally active infected cells[10–13]. We observed that the fraction of T$_{RM}$ expressing CD32 was enriched for most of the molecules investigated, many of them associated to HIV susceptibility. The fact that CD32 expression was strongly associated to cells expressing the highest levels of CD69 may explain why these cells are enriched for most of these molecules, since many are connected to the T$_{RM}$ phenotype definition *per se*[25]. We also observed an association between CD32 and CD69 expression in peripheral blood of these ART-suppressed patients, and CD32$^+$ CD69$^+$ CD4$^+$ T cells have already been shown to correlate with proviral DNA in blood[10]. Supporting their higher susceptibility to HIV infection in cervix, we detected remarkably high copies of HIV-1 DNA per cell associated to this fraction. Moreover, the overall level of tissue infection appeared to determine the level of infection of the CD32$^-$ T$_{RM}$ fraction, where less infection magnified the preference towards CD32 expressing T$_{RM}$. Further, productively infected cells expressed more CD32 than uninfected T$_{RM}$ and these cells were associated to higher expression of HLA-DR, as recently reported[10]. Therefore, CD32 expression in CD4$^+$ T$_{RM}$ could identify a subset remarkably susceptible to HIV-1 infection, potentially highly activated within an overall more quiescent T$_{RM}$ compartment, which may or may not continue to express this marker throughout the different stages of the retroviral infection. Future studies will confirm if CD32 is indeed predicted in the precursors of HIV-1 infected cells and if no down-regulation occurs in between HIV-1 infection stages. Our results, including the increased coexpression of IL-receptors in this subset, together with the fact that a recent paper demonstrates higher proportion of HIV-RNA$^+$ cells co-expressing CD32-RNA in ART-suppressed than in viremic individuals[13], highlight the need to better understand the dynamics of CD32 in the context of tissue reservoir establishment and persistence.

We found that HIV infection has long-term consequences for the CD4$^+$ T$_{RM}$ population of the cervical mucosa. In cervix from ART-suppressed women, the frequencies of CD4$^+$ and CD4$^+$ T$_{RM}$ cells out of the total CD45$^+$ lymphocytes decreased compared to tissues from uninfected women, and these cells expressed higher percentages of HLA-DR. Concomitant cervical viral infections such as herpes simplex virus type 2 (HSV-2) or human papilloma virus (HPV) have been associated with systemic T cell

activation[54,55], which could be conditioning our results. While Spain has low prevalence of HSV-2 infection compared to other countries and regions[56,57], six out of the 19 patients included in these analyses tested positive for HPV and presented cervical intraepithelial neoplasia grade II (CIN II). However, HPV and high-grade CIN on HIV$^+$ women are associated with high cervical CD4$^+$ cell density in both stroma and epithelium[54], and we did not detect any correlation between the studied frequencies and these conditions. On the other hand, it has been observed that menstrual blood of HIV-infected patients has lower expression of the integrin CD103 compared to uninfected women[42]; however, we did not detect a significant impact of HIV infection on the cervical CD4$^+$ CD103$^+$ T$_{RM}$ fraction in our patients. Although CD103 has been established as one of the core surface markers for the T$_{RM}$ lineage, CD4$^+$ T$_{RM}$ express lower levels of this integrin compared to the CD8$^+$ fraction[25], which may be related to higher intraepithelial retention of CD8$^+$ vs. CD4$^+$ T cells[47]. Still, detection of HIV-1 RNA in cervical tissue from women demonstrated intraepithelial and submucosal positive cells, highlighting the presence of these cells in close proximity to the mucosal surface, which has obvious implications for HIV acquisition. In this sense, Hladik et al. demonstrated that intraepithelial vaginal CD4$^+$ T cells, which highly expressed the major HIV-1 coreceptors (CD4 and CCR5), were remarkably permissive to HIV-1 infection[58]. It remains to be elucidated how long these cells remain within these mucosal tissues and if it they can establish a latent cellular reservoir.

Generation of CD4$^+$ T$_{RM}$ has been associated to protection against herpesvirus in the female genital tract of mice[48], and it is also believed to limit recurrence in humans[47]. In mice, protective CD4$^+$ T$_{RM}$ were found in macrophage-organized memory lymphocyte clusters that developed after vaginal but not systemic attenuated infection[48]. Protection against other sexually transmitted infections such as Chlamydia is also associated to presence of effective CD4$^+$ T$_{RM}$ in the genital mucosa[47,49]. Female reproductive tract conditions, such as bacterial vaginosis, induce an increase in the expression of α4β1 and CCR5 of circulating effector CD4$^+$ lymphocytes[59], which may correspond to recruited cells that contribute to the establishment of a memory pool. Remarkably, it is now clear that pre-existing T$_{RM}$ populations are not displaced after subsequent infections, which enables multiple T$_{RM}$ specificities to be stably maintained within the tissue, as recently shown in the skin[29]. Thus, in principle, infections over the lifetime of an individual will contribute to increase the T$_{RM}$ pool. Further, novel vaccine designs combining a general prime strategy with recruitment of activated T cells to the genital tract (by means of topical chemokine application) have been proposed to establish local memory and prevent reinfection[48]. However, in the context of HIV infection, establishment of these clusters of CD4$^+$ T$_{RM}$ may also provide increased number of HIV-1 targets for the nascent infection, which can ultimately contribute to a larger reservoir after treatment. In this sense, we detected CD69$^+$ enriched areas compatible with T$_{RM}$ clusters in the cervical submucosa of ART-suppressed patients that contained transcriptionally active HIV cells. Therefore, past and concomitant infections of the genital tract may contribute to lowering the threshold for HIV acquisition[60] by increasing preferential T$_{RM}$ targets and, in addition, augment long-lived cellular reservoirs.

The effect of cytokines on the maintenance and homeostatic proliferation of certain populations of memory lymphocytes may be determinant in the establishment and durability of reservoirs, as occurs for IL-7 driven homeostatic proliferation of latently infected cells[5] or for IL-15 with the T$_{SCM}$ subset[61]. It seems that the maintenance of effector memory cells after the contraction phase of the immune response corresponds to 5–10% of the effectors that express high levels of CD127[62]. Thus, it is possible

that the virus takes advantage of this population at the moment of contraction of the immune response and the transition to a resting state in order to establish latency in these cells. Further, while both CD127 and CD57-expressing cells in tonsils are permissive to HIV infection, only CD57$^+$ cells lead to productive infection[41]. According to detailed phenotyping of T$_{RM}$ in different tissues across the human body, maintenance of CD127 and low expression of CD57 (the latter at least in CD8$^+$ T$_{RM}$) are features of T$_{RM}$[25]. Additional lack of recent activation markers and reduced proliferative turnover shape T$_{RM}$ as quiescent long-lived cells designed for local immunity[25]. Here we have confirmed high expression of CD127 in CD4$^+$ T$_{RM}$ from the cervical mucosa, as well as increased expression of CD122 and CD132, with low expression of HLA-DR, suggesting that they indeed could constitute a long-term stable reservoir. Further, we also concur the lack of expression of CD127 among productively infected cells, thus it remains to be determined if this receptor can identify latently infected subpopulations of T$_{RM}$ in HIV patients. Consistent with this idea, it has been shown that stimulation with IL-7 induces reverse transcription and virus integration[63], while addition of this cytokine to the cervical explant model enhances HIV replication by promoting not only proliferation but also survival of infected cells[64]. The fact that cervical T$_{RM}$ express high levels of these receptors reinforces their potential contribution to HIV persistence in these tissues.

Importantly, regardless of long ART-mediated viral suppression, we detected very high levels of proviral DNA in the T$_{RM}$ fraction of cervical tissue from HIV$^+$ treated patients. The values of HIV-1 DNA copies detected here concur with recent data reporting higher provirus copy numbers found in the lamina propria of vaginal tissue of a single ART-treated woman compared to PBMC[21]. In addition, our results on proviral DNA are remarkably similar to the ones determined in ileum biopsies[16] and in bronchoalveolar lavage[17] from ART-suppressed patients, in which effector memory phenotypes were reported to sustain these reservoirs. Since most tissue T$_{RM}$ express phenotypic markers that would classify them as circulating memory effector cells, based on the expression of CD45RA and CCR7[26], it seems plausible that the aforementioned reservoirs are largely composed of T$_{RM}$ phenotypes. Still, in blood samples from ART-treated patients, most of the HIV-1 DNA detected contains fatal mutations and only 2–10% of proviruses are estimated to be intact, indicating that only a minor fraction is more likely to produce replication competent virus[65]. Moreover, we showed that after long-term ART, T$_{RM}$ contained transcriptionally active HIV. Although we performed a quantitative viral growth assay in sorted T$_{RM}$ obtained from cervix of one hysterectomy, the small number of cells recovered limited this assay. Future studies involving techniques with lower cell number requirement may better clarify the portion of intact virus within T$_{RM}$ from the cervix of ART-treated women[66]. However, the fact that a recent study reports that proviral copy numbers correlate with levels of cell-associated HIV RNA and predict control after treatment interruption[55] denotes the importance of the remaining levels of vDNA and vRNA encountered in this study. These findings recognize this subpopulation of cells, not only as a long-term cellular reservoir for HIV, but also as the possible source of residual viremia observed in HIV$^+$ patients despite ART. In this sense, a high degree of variability in penetration of the different antiretroviral drugs, both between and within drug classes, has been reported in the female genital tract[67], potentially contributing to this phenomenon, as observed in other tissues[3,4].

In summary, in this study we have identified, CD4$^+$ T$_{RM}$ as a critical cellular reservoir of the cervical tissue, and most likely of other tissues not yet identified. These cells are preferentially infected during ex vivo HIV infection and, after years of

suppressive treatment, still harbor HIV-1 nucleic acids in the cervical mucosa of infected women. With current clinical guidelines recommending early ART initiation, the relevance of mucosal tissues where the virus first replicates as sources of viral persistence may increase in years to come. Overall, the contribution of CD4$^+$ T$_{RM}$ to viral persistence in lymphoid and mucosal tissues as well as their capacity to establish latent or active viral reservoir in these tissues requires major attention in order to reach a functional cure in HIV infected patients.

## Methods

**Study subjects and patient samples.** Cervical tissue was obtained from non-neoplastic hysterectomies ranging 26–74 years performed in participating hospitals. Additionally, blood and cervical tissue from ART-treated HIV$^+$ women were obtained from women undergoing a hysterectomy or a cone biopsy in the same hospitals. Study protocols were approved by the corresponding Ethical Committees (Institutional Review Board numbers PR (IR)294/2017 for the HUVH, PI-17–159 for the HUGTP and 2018/8017/I for the Parc de Salut Mar), and written informed consent was provided by all patients recruited to this study. Patient information on plasma viral loads, CD4$^+$ T cell counts, time of ART-suppression and treatment is summarized in Table 1.

**Cervical tissue digestion and flow cytometry phenotyping.** A piece of ectocervix and endocervix from healthy donors was delivered to the laboratory once healthy tissue status was confirmed by the Pathology Service. Tissue was delivered in antibiotics-containing RPMI 1640 medium and processed within the 24 h following surgery[68]. The mucosal epithelium and the underlying stroma were separated from the muscular tissue and dissected into ~8-mm$^3$ blocks. Tissue blocks were then enzymatically digested by incubating them with 5 mg/ml collagenase IV in RPMI 1640 5% FBS for 30 min at 37 °C and 400 rpm, followed by a mechanical dissociation of the tissue with a disposable pellet pestle[68]. Once digested, the cellular suspension was washed twice with PBS and stained with Live/Dead Aqua (Invitrogen) to identify dead cells. After washing, cells were equally divided into 5 different tubes and stained with the following antibodies: HLA-DR-PerCP-Cy5.5 (G46-6 dilution 1/100, cat. no. 560652), CD14-V450 (MØP9, dilution 1/40, cat. no. 560349), CD69-Horizon-PE-CF594 (FN50, dilution 1/20, cat. no. 562645), CCR5-APC-Cy7 (2D7, dilution 1/50, cat. no. 557755), CD163-BV786 (GHI/61, dilution 1/20, cat. no. 741003), CD4-BV605 (RPA-T4, dilution 1/12.5, cat. no. 562658), CD19-V500 (HIB19, dilution 1/50, cat. no. 561121), CCR7-PE (3D12, dilution 1/10, cat. no. 552176), CD49d-FiTC (9F10, dilution 1/25, cat. no. 560840), β7-APC (FIB504, dilution 1/50, cat. no. 551082), CD184-APC (12G5, dilution 1/40, cat. no. 555976) (all from BD Biosciences), CD3-SuperBright 645 (OKT3, dilution 1/20, cat. no. 64-0037-42), S1PR1 (CD363)-eFluor660 (SW4GYPP, dilution 1/72, cat. no. 50-3639-42) (eBiosciences), CD45-Alexa Fluor 700 (HI30, dilution 1/100, cat. no. 304024), CD32-PE-Cy7 (FUN-2, dilution 1/50, cat. no. 303214), CD103-FiTC (BER-ACT8, dilution 1/50, cat. no. 350203), CD49a-PE (TS2/7, dilution 1/200, cat. no. 328303), CXCR3-FiTC (G025H7, dilution 1/33.3, cat. no. 353703), CD127-FiTC (A019D5, dilution 1/100, cat. no. 351312), CD132-PE (VI C-89, dilution 1/100, cat. no. 338606), CD122-APC (TU27, dilution 1/100, cat. no. 339008), CCR6-APC (G034E3, dilution 1/100, cat. no. 353416) (all from BioLegend), CXCR6-APC (56811, dilution 1/25, cat. no. FAB699A), CCR2-PE (48607, dilution 1/33.3, cat. no. FAB151P) (R&D Systems), TCR-γδ-FiTC (11F2, dilution 1/40, cat. no. 130-096-884) and CD161-PE (191B8, dilution 1/50, cat. no. 130-092-677) (Miltenyi Biotec). Staining of T-bet (T-bet-BV421, 4B10, dilution 1/40, cat. no. 644815, Biolegend), Eomes (Eomes-PE-Cy7, WD1928, dilution 1/33.3, cat. no. 25-4877-42, eBioscience) and Hobit (Hobit-AF647, Sanquin-Hobit/1, dilution 1/20, cat. no. 566250, BD Biosciences) was performed using FoxP3 transcription factor staining buffer set (eBioscience), according to manufacturer's protocols. Preliminary controls on the effect of collagenase IV treatment on the stability of the cell surface markers analyzed were performed in PBMC[68]. Fluorescence Minus One (FMO) controls were performed to delineate gates of continuous markers in at least one tissue sample. Samples were acquired in a BD LSRFortessa flow cytometer and analyzed with FlowJo vX.0.7 software (TreeStar).

**Flow cytometry imaging of CD32 expressing cervical T$_{RM}$.** Specific analysis of CD32 expression on cervical CD4$^+$ cells was performed by using Amnis® technology. To do so, digested tissue cells were stained with CD32-PE (FUN-2 dilution 1/20, cat. no. 303205), CD45-Alexa Fluor 700 (HI30, dilution 1/100, cat. no. 304024, both from BioLegend), CD69-FITC (FN50, dilution 1/20, cat. no. 557049), CD4-BV605 (RPA-T4, dilution 1/12.5, cat. no. 562658 both from BD Biosciences) and a viability dye (LIVE/DEAD Fixable Violet Dead Cell Stain Kit, dilution 1/250, cat. no. L34966, Invitrogen), and then acquired with an AMNIS ImageStremX imaging flow cytometer (Merck). Data were analyzed using IDEAS v6.1 software. Gradient RMS value > 40 was established as a threshold for best focus, and only focused cells were considered for analysis. A CD32-FMO control was performed to delineate this gate in CD69$^+$ and CD69$^-$ cervical CD4$^+$ cells.

**Explant infection and subset purification.** Eight to twelve blocks of tissue per condition were placed in RPMI1640 media supplemented with 20% FBS (R20) in a 24-well plate and infected with 7200 TCID$_{50}$ of the viral stock HIV-1$_{BaL}$ or medium in control wells. For each experimental condition at least duplicates were performed. After 2 h of incubation at 37 °C, tissue blocks were washed three times with 3 mL of PBS in 6-well plates and placed back into a 12-well plate at 8 blocks/well in 1 mL of R20. Infected cervical tissue blocks and controls were cultured for additional 10–12 days, with a change of medium every 3 days. Tissue digestion with collagenase IV (Invitrogen) and manual dissociation were immediately executed, as described above[68]. Supernatants and cell suspensions were incubated with Aqua vital dye and stained with CD69-Horizon-PE-CF594 (FN50, dilution 1/20, cat. no. 562645), CD4 or CD127-BV605 (RPA-T4, dilution 1/12.5, cat. no. 562658 and A019D5, dilution 1/100, cat. no. 351333 respectively), CD19-V500 (HIB19, dilution 1/50, cat. no. 561121), CD3-PerCP (SK7, dilution 1/10, cat. no. 345766), CD14-APC-H7 (MφP9, dilution 1/66.7, cat. no. 560180), CD8-APC (RPA-T8, dilution 1/40, cat. no. 555369), HLA-DR-BV421 (G46-6, dilution 1/200, cat. no. 562804) (all from BD Biosciences), CD49a-APC-Vio770 (REA1106, dilution 1/50, cat. no. 130-119-410, Miltenyi Biotec), CD45-Alexa Fluor 700 (HI30, dilution 1/100, cat. no. 304024), CD32-PE-Cy7 (FUN-2, dilution 1/50, cat. no. 303214) and CD103-FiTC (BER-ACT8, dilution 1/50, cat. no. 350203) (BioLegend). After surface staining cells were intracellularly stained with KC57-PE antibody (FH190-1-1, dilution 1/40, cat. no. 6604667, Beckman Coulter, Brea, CA) using Fixation & Cell Permeabilization Kit (Invitrogen) according to manufacturer's instructions to determine p24 expression[10]. 1% PFA fixed samples were acquired in a BD FACSAria™ II Cell Sorter for purification of Aqua$^-$CD19$^-$CD45$^+$ CD3$^+$ CD8$^-$CD4$^\pm$ T cells: 1) T$_{RM}$ as CD69$^+$; 2) non-T$_{RM}$ as CD69$^-$ and 3) CD32$^+$ T$_{RM}$ as CD69$^+$CD32$^+$. These samples were immediately lysed and stored for posterior vDNA analyses by qPCR as follows. Further, samples were acquired for phenotype analysis using FlowJo vX.0.7 software.

**Cervical T$_{RM}$ purification and infection.** Cervical tissue from healthy donors was obtained and digested, and the resulting cell suspension was stained for viability with Aqua vital dye (Invitrogen) and the following antibodies: CD69-FiTC (FN50 dilution 1/20, cat. no. 557049), CD45-PE (HI30, dilution 1/100, cat. no. 560975), CD3-PerCP (SK7, dilution 1/10, cat. no. 345766), CD19-V500 (HIB19, dilution 1/50, cat. no. 561121), PD1-BV421 (EH12.1, dilution 1/20, cat. no. 562516) (all from BD Biosciences), CD4-APC (OKT4 BioLegend, dilution 1/80, cat. no. 300514). Aqua$^-$CD19$^-$CD45$^+$ CD3$^+$ CD4$^+$ T cells were acquired in a BD FACSAria™ II Cell Sorter and separated according to their CD69 expression into CD69$^+$ (T$_{RM}$) and CD69$^-$ (non-T$_{RM}$). Sorted cells were then infected with 3655 TCID$_{50}$ of the viral stock HIV-1$_{BaL}$ for 4 h at 37 °C or with medium in control conditions, washed with PBS and cultured in a 96-wells plate. After three days, cells were stained for viability with Live/Dead Fixable Far Red (Invitrogen), CD69-FiTC (FN50 dilution 1/20, cat. no. 557049) and HLA-DR-PerCP-Cy5.5 (G46-6, dilution 1/100, cat. no. 560652), before staining for p24 as described above. Fixed cells were finally acquired in a BD FACS Calibur flow cytometer and analyzed with FlowJo vX.0.7 software (TreeStar).

**Cervical T$_{RM}$ migration assay.** Nine blocks of tissue per experimental condition were infected and cultured for 10 days as described above. On day 10, tissue was transferred to a transwell insert containing CCL19 (100 ng/ml, Biolegend), CCL21 (100 ng/ml, R&D) and sphingosine-1-phosphate (S1P, 10 μM, Sigma) in the lower chamber medium (R20). Triplicates were established for each experimental condition, as well as the corresponding controls. Tissue was left overnight at 37 °C and then digested as described above. Migrating and tissue resident cells were stained separately for viability, surface markers listed above and p24, fixed, acquired in a BD FACSAria™ II Cell Sorter and analyzed with FlowJo vX.0.7 software (TreeStar).

**Sample subset purification from ART-treated HIV$^+$ patients.** Samples from blood and cervical tissue from HIV-infected women under suppressive ART were processed to obtain PBMC by Ficoll–Paque density gradient centrifugation or a cell suspension from digested tissue, as described above. Cells were surface stained with the same panel described for the ex vivo infection, without p24 intracellular staining. Fixed cells were isolated by cell sorting using a BD FACSAria™ II Cell Sorter. In both tissues sorted cells were Aqua$^-$CD19$^-$CD45$^+$ CD3$^+$ CD4$^\pm$CD8$^-$T cells and, depending on the tissue, different subsets were purified: for cervix (1) T$_{RM}$ were CD69$^+$ and non-T$_{RM}$ were CD69$^-$; for blood (1) CD69$^+$ (CD103$^-$ CD32$^{+/-}$), (2) CD103$^+$ (CD69$^-$ CD32$^{+/-}$), (3) CD32$^+$(CD69$^-$ CD103$^-$) and (4) CD4$^{+/-}$ T cells did not express any of these three markers (CD69$^-$ CD103$^-$ CD32$^-$). These samples were immediately lysed and stored for posterior vDNA analyses by qPCR as follows. Further, samples were acquired for phenotype analysis using FlowJo vX.0.7 software.

**Quantification of cell-associated HIV-1 DNA by qPCR.** All sorted subsets obtained from the ex vivo cervical model or from blood and cervix of HIV$^+$ patients were immediately lysed with a proteinase K-containing lysis buffer (55 °C over-night and afterwards 5 min at 95 °C) and cell lysates were subjected to total HIV-1 DNA quantification by qPCR with primers against HIV long terminal repeat (LTR forward 5′-TTAAGCCTCAATAAAGCTTGCC-3′ and LTR reverse 5′-GTTCGGGCGCCACTGCTAG-3′; LTR probe 5′-CCAGAGTCACACACCA GACGGGCA-3′). For quantification of total HIV-1 DNA, a standard curve was prepared and CCR5 gene was used for cell input normalization (CCR5 forward 5′-

GCTGTGTTTGCGTCTCTCCCAGGA-3′ and CCR5 reverse 5′-CTCACAGCCC TGTGCCTCTTCTTC-3′; CCR5 probe 5′-AGCAGCGGCAGGACCAGCCCCA AG-3′). The cycling parameters were 50 °C for 2 min, 95 °C for 10 min and then 95 °C for 15 s and 60 °C for 1 min for 55 cycles of amplification. Contribution of each subset to the HIV reservoir was calculated by considering the frequency of each subset (analyzed by flow cytometry) within the total CD4 compartment and their relative infection frequency.

**RNA in situ hybridization and immunohistochemistry**. Paraffin-embedded cervical tissue samples from one viremic and six aviremic ART-treated HIV-infected patients were obtained from the Pathology Department of the HUVH. For detection of RNA, we used ultrasensitive RNA detection assay RNAscope 2.5 HD Duplex Reagent Kit (Anacrome). A high sensitivity target-specific probes to GagPol HIV mRNA sequence were used (bases 507–4601; reference #31769146). Cervical tissue sections of 5 μm were mounted on Superfrost Plus microscope slides (Fisher Scientific). The assay was performed according to manufacture's instructions. Briefly, samples were deparaffinized with xylene and dehydrated in 100% ethanol. Heat-induced epitope retrieval and protease digestion were used as sample pre-treatment. Next, probes were incubated for 2 h at 40 °C and samples were stored overnight in 5× saline sodium citrate buffer. Next day, amplification and signal development of RNA probes were performed by sequential incubation of indicated reagents. Afterwards, an immunohistochemisty protocol was carried out. Samples were first blocked by incubation with 1× Tris-buffered saline buffer containing 1% bovine serum albumin and 10% normal donkey serum for 30 min. Then, anti-CD69 antibody (1/100, Abcam) was incubated overnight at 4 °C. Antibody detection was performed with DAB detection kit (Abcam), according to manufacture's instructions. Images were captured using a DP71 digital camera (Olympus, Center Valley, PA, USA) attached to a BX41 microscope (Olympus). Specificity of HIV-RNA positive cells was determined by quantifying the relative false detection rate in cervical tissues from two HIV negative patients, which corresponded to 0.03 positive cells/mm², with no positive cell detected in their epitheliums or in CD69 enriched areas (Supplementary Fig. 5). Additional negative controls consisted of a matched sample without target-specific probes and a control without primary antibody (CD69) for immunohistochemistry (Supplementary Fig. 5). A cell was considered HIV RNA positive based on the diameter of the staining, while small punctate signals were not considered in the quantification[46]. Positive cells within the whole tissue section were counted by visual inspection and the total tissue area was measured by ImageJ software.

**Statistical analyses**. All the data were analyzed using the software GraphPad Prism 7.0 (GraphPad Software, La Jolla, CA, USA). All values in the graphs are expressed as the median and the range and the interquartile range. Non-parametric two-tailed Mann–Whitney rank tests, Wilcoxon matched-pairs signed rank test and Friedman test with Dunn's correction for multiple comparisons were used. A $P$ value < 0.05 was considered significant.

**Reporting summary**. Further information on research design is available in the Nature Research Reporting Summary linked to this article.

## Data availability
The source data underlying both the Main and Supplementary Figures are provided as a Source Data file. All other data are available from the corresponding authors upon reasonable requests.

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

## Acknowledgements

We would like to thank all the patients who participated in the study and their providers. We thank José L. Poza, Mª Elena Suárez, Mª Assumpció Pérez-Benavente and Cristina Carrato for referral of patients and sample collection, Irian Lorencés and Laia Pérez-Roca from the Tumor Bank of the IGTP-HUGTiP for sample management, Alba Ruiz and Ruth Peña for generating the viral stock R5-Bal and Gerard Requena from the Flow Cytometry Facility at the IGTP for excellent technical assistance, as well as Isabel Crespo from the Flow Cytometry Platform at the IDIBAPS for her excellent technical assistance on Amnis technology. This work was primarily supported by grants from the Spanish "Ministerio de Economía y Competitividad, Instituto de Salud Carlos III" (ISCIII, PI14/ 01235 and PI17/01470) and a fellowship award from the Dexeus foundation for women's health research to M.G., grants R21AI118411 and SAF2015-67334-R (from the Spanish Secretariat of Science and Innovation and FEDER funds) to M.J.B and grants from the ISCIII (PI14/01058 and PI17/00164) to J.G.P. M.G., M.J.B., and. J.G.P. are supported by the Spanish AIDS network Red Temática Cooperativa de Investigación en SIDA (RD16/ 0025/0007) M.G. is currently supported by the "Pla estratègic de recerca i innovació en salut" (PERIS, SLT002/16/00353), from the Catalan government, while the Miguel Servet program from the ISCIII supports M.J.B. (CP17/00179) and J.G.P. (CP15/00014).

## Author contributions

J.C.-P., J.G.-E., C.S.-P., D.A.R., L.L.-B., A.A.-G., M.A.F., J.G.P., M.J.B., and M.G. designed and carried out experiments. J.C.-P., J.G.-E., C.S.-P., D.A.R., M.J.B., and M.G. analyzed and interpreted data. J.C., T.S., G.T., B.LL., J.M.S.-S., A.T., C.L., L.M.-B., C.C.-M. and V.F. selected study subjects and provided samples. J.C.-P., M.J.B. and M.G. drafted the manuscript and all authors edited the final version of the manuscript.

## Competing interests

The authors declare no competing interests.
