## [Peer Review File · Nature Communications]

Reviewers' comments:

Reviewer #1 (Remarks to the Author):

This article focuses on characterizing the HIV latent reservoir in the cervix, particularly the contribution of CD4+ T resident memory cells. The authors first conducted some FACS-based phenotyping to characterize to what extent various antigens are co-expressed with non-Trm (CD69-) or Trm (CD69+) cells from the cervix of uninfected individuals, and further separated the Trm population into those expressing CD32 or not. They then conducted HIV infection assays in cervical explants and characterized cells that were productively infected with HIV. Finally, they procured cervical explants isolated from HIV-infected women and quantitated HIV DNA levels in sorted populations of cells, and also conducted IHC to examine co-expression of CD69 with HIV RNA. Although understanding the nature of the latent reservoir in the female reproductive tract is an extremely important area of research, the experiments presented in this study fall short of advancing our knowledge of the nature of this reservoir. Furthermore, important controls are missing, and the conclusions drawn are not always supported by the data. Specific comments are presented below:

1. It is not clear why there is such a focus on analyzing CD32 expression on CD4 T cells. The authors failed to discuss the notion that CD32 is no longer thought to be a marker of latent cells, and reports that CD32 expression on CD4 T cells could be artifactual (PMIDs 30232425, 30232423). It's not clear what meaningful conclusion the authors would like the reader to draw from their analysis of CD32+ vs. CD32- cells, and to what extent the authors confirmed that what they are characterizing is not an artifact of FACS as recently suggested.
2. The gating strategies in Fig. 1A and 2A are poorly described. It's not clear what gate is leading to what population of cells. A viability dye does not seem to have been used (e.g., Fig. 2A) which is highly concerning given the known poor viability of cervical lymphocytes and the fact that dying cells are non-specifically sticky for antibodies during FACS staining. Fig. 2A is extremely confusing and not labeled. E.g., which gate leads to which population, and which samples are uninfected vs. infected.
3. In Fig. 1B-L, it's not clear what the data are gated on. Also, instead of examining the % of each population expressing the indicated markers, a more meaningful analysis would be to what extent these various markers are co-expressed (especially for the IL2, IL15, and IL7 heterodimeric receptors).

4. An extremely important caveat of the ex vivo infection experiment, which the authors themselves acknowledge, is that they have not ruled out the possibility that HIV upregulates CD69 upon infection, which it could very well do since HIV infection can promote T cell activation and CD69 is an activation marker. Without addressing this issue, the authors cannot make a claim that Trm are preferentially infected ex vivo. That being the case, it is not clear what meaningful conclusion is to be drawn from the ex vivo infection experiments. Furthermore, the ex vivo infection experiments are looking only at productive infection, whereas the manuscript focuses supposedly on the latent reservoir. Are the authors suggesting that the latent reservoir in the cervix of ART-suppressed women are all productively-infected cells?
5. The purpose the analysis in Fig. 3 is not clear.
6. [redacted]
7. In Fig. 4, the authors acknowledge that the majority of CD4+ T cells in their biopsy samples are Trm; if we go by the gates shown in Fig. 2A, >80% of the CD4+ T cells are defined as Trm. That being the case, if very few non-Trm are sorted out, sampling bias could account for their inability to detect HIV DNA in those cells. This is particularly of concern since all but one sample had more sorted Trm than non-Trm, and it was hinted that no HIV DNA was detected in any of the non-Trm that were sorted at low numbers.
8. Why in Fig. 4B are the data presented as total CD4+ T cells vs. CD4+ Trm, when the sorted cells were CD4+ Trm vs. CD4+ non-Trm. Is it because most in almost all the non-Trm samples no HIV DNA was detectable?
9. Fig. 5 describing the IHC experiments lack negative controls (e.g., staining from uninfected individuals, IHC controls, etc). It is not clear why in Fig. 5B the two representative samples (one on the left and one on the right) look so different. The one on the left looks like there is minimal CD69 staining, while the one on the right it looks like every cell has CD69 (background?) staining.
10. The FMOs in Fig. S1 are not appropriate, as there is significant positivity in the FMO gates. There are no x- and y-axis labels.
11. Fig. S2A is lacking the important control of an uninfected culture taken through the same timecourse.

Reviewer #2 (Remarks to the Author):

In this paper, the authors aimed to study the HIV reservoir in cervical tissues. A major focus was to understand if tissue resident memory (TRM) CD4+ T cells are infected with HIV and represent a sizeable proportion of the reservoir. The authors found that the majority of CD4+ T cells in cervix expressed CD69, a canonical marker of TRMs, and demonstrate that many molecules associated with HIV susceptibility is expressed by TRMs and CD32+ TRMs. Quite remarkably, the authors found >200 times more HIV-proviral copies in cervical than blood CD4+ T cells. A vast majority of the total infected CD4+ T cells demonstrated a TRM phenotype.

This is an interesting paper for the HIV field and highlight that the reservoir composition in tissues is remarkable different from blood. It is the first data directly examining if TRMs are primary targets of HIV infection and viral persistence. HIV likely persist to a very high degree in TRMs given that the vast majority of lentiviral-infected cells are present in mucosal tissues (Estes et al, 2018, Nat Med). Given that a recent publication in mice found that CD4+ T cells in the female reproductive tract are essentially all TRMs (Beura et al, 2019, JEM; this publication should be cited), it altogether would make sense that TRMs are dominating in cervix also in humans. However, there are a couple of more experiments that could be done in order to truly validate this finding.

Major comments:

CD69 is a valid marker of TRMs, but not all CD69+ cells might be bona fide TRMs. In light of recent data from the Masopust lab and others (Beura et al, 2018, Immunity etc), it seems rationale to believe that CD62L-/CCR7-S1PR1- CD69+ CD4+ T cells are the true TRMs in tissues. The authors do provide data that most CD69+ CD4+ T cells do not express CCR7 in the first supplementary figure, but also other egress receptors (such as S1PR1) and transcription factors (KLF2, KLF3 etc) are usually low in TRMs. Furthermore, TRMs usually have a specific gene signature that includes upregulation of specific chemokine receptors and transcription factors (for instance Hobit, Blimp-1 etc). Have the authors conducted RNAseq analysis or gene-expression analysis to confirm that the CD69+ CD4+ T cell signature in cervix is similar to a bona fide TRM signature published by others (for instance Kumar et al, 2017, Cell Reports)?

The authors treat infected blocks with CCL19 and CCL21 to assess the degree of infected cells in supernatant. Why did the authors try to use these chemokines given that CCR7 is low on both CD69- and CD69+ CD4+ T cells in the cervix? The assay does not seem to work given that the number of infected cells is similar between the CCL19/CCL21 treated and untreated condition. The authors should if anything instead then use S1P in these assays to lure out S1PR1+ T cells – which should be the CD69- CD4+ T cells.

Furthermore, despite more cells in the supernatant are CD69-, there is also a heavy skewing of infected cells being more CD69+ in the supernatant. Based on these data, it is not possible then to conclude that migrating infected CD4+ T cells have a circulating CD69- phenotype. The authors could

instead try a trans well migration assay and supplement with CCL19, CCL21 and S1P to determine if infected cells are moving towards a gradient in the lower wells. This is the most established way in the field to provide some evidence of migration capacity.

[redacted]

Why do the authors only gate for CD32dim cells in their data set? The initial Nature paper also included CD32hi cells.

In the field there is a debate on whether HIV replication is ongoing or not after ART. The authors found >200 times more proviral DNA copies per cell in cervix in comparison to blood – this is a remarkable number. Despite that, the authors claim that few TRMs are activated. There are literature suggesting that TRMs are in a pseudo-activated state and poised to rapidly exert effector functions following antigen recognition. The authors also demonstrate more HLA-DR+ cells within the CD4 compartment in HIV+ subjects despite ART, which could suggest that some cells are activated and ongoing viral replication might take place despite ART.

Is there any evidence of more proviral copies within the CD69+HLA-DR+ compartment in cervix? [redacted]. Could it be poor penetration of drugs in the cervix explaining the higher HIV-DNA levels in this site, similar to lymphoid tissues?

Minor comments:

Overall, the authors need to reformat many of the flow plots. It seems like many flow plots have been directly transported from flowjo and not really formatted in illustrator or another program to remove unnecessary gating data. Be consistent at least and label the X- and Y-axis with the markers and fluorophores in the same way for all plots and remove data within the gates except for the percentage. Make the X- and Y-legends for each marker larger (similar to Figure S2) and keep the labeling consistent. Avoid overlap of multiple gates over each other if possible as it becomes hard to visualize the data.

Reviewers' comments:

Reviewer #1 (Remarks to the Author):

Summary. This article focuses on characterizing the HIV latent reservoir in the cervix, particularly the contribution of CD4⁺ T resident memory cells. The authors first conducted some FACS-based phenotyping to characterize to what extent various antigens are co-expressed with non-Trm (CD69⁻) or Trm (CD69⁺) cells from the cervix of uninfected individuals, and further separated the Trm population into those expressing CD32 or not. They then conducted HIV infection assays in cervical explants and characterized cells that were productively infected with HIV. Finally, they procured cervical explants isolated from HIV-infected women and quantitated HIV DNA levels in sorted populations of cells, and also conducted IHC to examine co-expression of CD69 with HIV RNA. Although understanding the nature of the latent reservoir in the female reproductive tract is an extremely important area of research, the experiments presented in this study fall short of advancing our knowledge of the nature of this reservoir. Furthermore, important controls are missing, and the conclusions drawn are not always supported by the data. Specific comments are presented below:

1. It is not clear why there is such a focus on analyzing CD32 expression on CD4 T cells. The authors failed to discuss the notion that CD32 is no longer thought to be a marker of latent cells, and reports that CD32 expression on CD4 T cells could be artifactual (PMIDs 30232425, 30232423). It's not clear what meaningful conclusion the authors would like the reader to draw from their analysis of CD32⁺ vs. CD32⁻ cells, and to what extent the authors confirmed that what they are characterizing is not an artifact of FACS as recently suggested.

We understand the reviewer's concern on the conclusions to be drawn from our analyses on CD32 expressing T_{RM}, as well as on the necessity to better clarify the purpose of its inclusion in these analyses.

First of all, to assure that CD4 T_{RM} expressing dim levels of CD32, as shown in the gating strategy (new Fig. 2a), are truly individual CD4⁺ T cells expressing this molecule with no contamination from cell doublets or conjugates with B cells or other subsets; we have now performed three independent experiments using Amnis-imaging FACS technology to determine the purity of these cells (new Sup. Fig. S1b and S1c). For these experiments cervical cell suspensions were stained with a viability dye and antibodies against CD45, CD4, CD69 and CD32 (as now stated in the Methods section). Each individual experiment included a CD32-FMO control to draw the CD32 gate in CD69⁺ or CD69⁻ cells (as shown in representative plots in new Sup. Fig. S1b). Using this technology and gating strategy, we unequivocally show individual cells expressing dim levels of CD32 associated to the T_{RM} fraction (CD69⁺) in a gate based on the FMO of the same sample, but not in the CD69⁻ fraction. These cells represent viable single cells that have surface expression of CD45, CD4, CD69 and CD32. Of note, in about 40 to 50% of the cells we detected some punctate staining, suggesting potential events of trogocytosis, although in some of these cases membrane staining was also detected (Sup Fig. S1c). Note that performing different controls

out of the same cervical tissue sample limits the amount of cells analyzed; yet the median frequency detected for CD32dim in CD4 T cells corresponds to the range detected by flow cytometry. This methodology has now been included in the material and methods section (lines 635-643) and results (lines 109-125).

Moreover, we have now contextualized these results with the references mentioned by this reviewer (Osuna et al. *Nature*, 2018; Bertagnolli et al. *Nature*, 2018; Perez et al. *Nature*, 2018), which have now been cited. Notice that the Gating Strategy that we used in the original manuscript (new Fig. 2a) was already very stringent in order to avoid T–B cell doublets (reported by Pérez et al. *Nature*, 2018 and Osuna et al. *Nature*, 2018). We have made an effort to better show the flow of this gating strategy as well as the exclusion of high CD32 cells (dotted black line) in this gating strategy. Finally, a new report demonstrates high proportion of HIV-RNA⁺ cells co-expressing CD32-RNA in gut CD3⁺ of ART-suppressed individuals (Vásquez et al. *Pathogens and Immunity*, 2019), supporting the existence of these cells in mucosal tissues. This reference has now also been cited.

In the study by Abdel-Mohsen M et al. (*Sci Trans Med*, 2018), in which we contributed with analyses of CD32 expression in HIV-infected CD4⁺ T cells from cervical tissue, we observed that there was an enrichment of CD32 among productively infected cells (p24⁺), which related to the overall message of the study, highlighting CD32 as marker of transcriptionally-active HIV⁺ cells (and not of the latent reservoir). This was already mentioned in the original manuscript. Many others, including the aforementioned paper from Vásquez et al., have now reinforced this message. In light of these results and, considering that we observed that cells expressing CD32 were mainly associated to the CD69⁺ T_{RM} fraction in cervical tissue (and not to the CD69⁻ non-T_{RM}), we decided to specifically address the characteristics of this T_{RM} fraction. Besides better clarifying the nature of these cells in the results section, we have also revised the discussion related to this marker to clarify the overall message (lines 470-493).

2. The gating strategies in Fig. 1A and 2A are poorly described. It's not clear what gate is leading to what population of cells. A viability dye does not seem to have been used (e.g., Fig. 2A) which is highly concerning given the known poor viability of cervical lymphocytes and the fact that dying cells are non-specifically sticky for antibodies during FACS staining. Fig. 2A is extremely confusing and not labeled. E.g., which gate leads to which population, and which samples are uninfected vs. infected.

We apologize for the low clarity of the gating strategies shown in Figs 1A and 1B (which now correspond to new Fig.1a/2a and Fig. 3a, respectively). We have now modified these figures to improve the understanding of the flow in the plots shown, as well as the subsets analyzed. Additionally, and considering the opinion of all the reviewers, we have reformatted all the plots considering consistency in the esthetics, but also complying with the journal requirements in terms of plot formatting (i.e. including axes scales and numbers).

Importantly, in every analysis performed in this study a viability dye was included (Live/Dead Aqua from Invitrogen, see material and methods). Each sample has its own frequency of cell death and the fact that the “lymphocyte” gate is already restricted in the gating strategy employed, may preclude the visualization of a high percentage of cell death. We hope that the new Figure formatting and corresponding Figure legends and results are now more clear and self-explanatory.

3. In Fig. 1B-L, it's not clear what the data are gated on. Also, instead of examining the % of each population expressing the indicated markers, a more meaningful analysis would be to what extent these various markers are co-expressed (especially for the IL2, IL15, and IL7 heterodimeric receptors).

We have modified the gating strategy in Fig.1A (now Fig. 2a), which now has a color code that matches the subset analyzed in each of the 1B to 1L graphs. This color-coding will certainly aid clarifying the subsets analyzed in the corresponding graphs.

Regarding to the co-expression of interleukin receptors, we now include this data in the results section (lines 168-172 and new Fig. S2b and c). As shown, non- T_{RM} had a small but significant fraction of cells co-expressing these receptors, while T_{RM} $CD32^+$ showed even higher frequencies compared to T_{RM} $CD32^-$. This is the case for $CD132$ and $CD122$ co-expression (which would form IL-2 and IL-15 receptors) and for $CD127$ and $CD132$ (which would form the IL-7 receptor). These results may indicate that, although in comparison with non- T_{RM} the frequency of each IL-chain individually is remarkably higher in T_{RM} regardless of their $CD32$ expression, as shown in the original manuscript (new Fig. 2j-l); indeed, a fraction of T_{RM} expressing $CD32$ may have more capacity to rapidly proliferate in response to these IL, which may be linked to the overall higher activation status of $CD32$ expressing T_{RM} . A comment about this has now been added in the discussion (lines 489-493).

4. An extremely important caveat of the ex vivo infection experiment, which the authors themselves acknowledge, is that they have not ruled out the possibility that HIV upregulates CD69 upon infection, which it could very well due since HIV infection can promote T cell activation and CD69 is an activation marker. Without addressing this issue, the authors cannot make a claim that Trm are preferentially infected ex vivo. That being the case, it is not clear what meaningful conclusion is to be drawn from the ex vivo infection experiments. Furthermore, the ex vivo infection experiments are looking only at productive infection, whereas the manuscript focuses supposedly on the latent reservoir. Are the authors suggesting that the latent reservoir in the cervix of ART-suppressed women are all productively-infected cells?

We agree on the fact that potential up-regulation of $CD69$ expression by HIV infection *per se*, is the most critical interference to claim the role of T_{RM} as preferentially-infected and long-term HIV-1 reservoir. We already acknowledge this issue in the original manuscript, and we performed several experiments to support our claims. We believe that 1) the dynamics of the expression of $CD69$ and HLA-DR during infection overtime, 2) the infection level observed in $CD103^+$ T_{RM} , and 3) the fact that most of the productive infection is retained within the tissue even after chemokine stimulation all together strongly suggest an important role for T_{RM} supporting HIV infection (new Figure 3 and Figure S3). However, and in order to better show the contribution of T_{RM} cells to HIV infection, we have now performed an additional experiment in which we sorted $CD69^+$ (T_{RM}) and $CD69^-$ (non- T_{RM}) $CD4^+$ T cells from freshly digested cervical tissue, which we then infected separately and determined the level of p24 antigen 3 days after. This experiment was performed two times in two separate donors each time (n=4; Fig. S3b). This experiment demonstrated that, as originally reported by Joag et al. (Mucosal immunology, 2016), cervical $CD69^+$ are preferential targets for HIV. We detected different percentages of p24 within $CD69^+$ (T_{RM}) but not in $CD69^-$ (non- T_{RM}) $CD4^+$ T cells, as stated now in the results section. Moreover,

while CD69⁺ (T_{RM}) partially maintained CD69 expression (even in the p24⁺ cells), we did not detect up-regulation of CD69 in the non-T_{RM} fraction even after exposure to HIV. These data has now been included in the results section (lines 231-237) and the corresponding methodology detailed (lines 666-677).

The purpose of studying T_{RM} infection in the *ex vivo* model was to determine the capacity of these cells to support productive infection in comparison to non-T_{RM} cells. Since we hypothesized that this cell fraction represents an important cellular reservoir in tissues, we rationalized that we first needed to address the susceptibility of these cells to HIV infection. Importantly, we observe that T_{RM} are preferentially infected compared with non-T_{RM} cells.

Regarding the last question raised by the referee referring to the nature of the reservoir in the cervix of ART-suppressed women, we assume that the HIV reservoir in this compartment may have different states of activity, ranging from truly “silent” latently-infected cells that are not actively transcribing, to transcriptionally-active but not producing virus, to productively infected cells (as suggested by the high levels of RNA detected tissue, Fig. 5). Although we cannot quantify each of these fractions, we have strong evidence supporting active transcription in the cervix of these ART-suppressed women. However, the *ex-vivo* model is used to address the infectivity of T_{RM} and the dynamics of CD69 expression (and other markers such as HLA-DR) during infection in this tissue, but it can obviously not recapitulate the impact that years of HIV-infection and ART therapy would have on this cellular compartment.

5. The purpose the analysis in Fig. 3 is not clear.

The purpose of Fig. 3 (new Fig.4) was to evaluate the impact of chronic HIV infection on the frequency of cervical CD4⁺ T_{RM}, since the *ex vivo* model can only provide information about short-term exposure to HIV, but not the overall effect in women infected and treated for years. Moreover, a previous report studying menstrual blood from HIV⁺ patients found low expression of CD103 (Moylan *et al. Pathogens & immunity, 2016*), which could be related to a depletion of the T_{RM} compartment. To determine the effect of HIV⁺ in this compartment, we compared different populations of CD4⁺ T cells in cervical tissue from HIV-infected and uninfected women.

6. [redacted]

7. In Fig. 4, the authors acknowledge that the majority of CD4+ T cells in their biopsy samples are Trm; if we go by the gates shown in Fig. 2A, >80% of the CD4+ T cells are defined as Trm. That being the case, if very few non-Trm are sorted out, sampling bias could account for their inability to detect HIV DNA in those cells. This is particularly of concern since all but one sample had more sorted Trm than non-Trm, and it was hinted that no HIV DNA was detected in any of the non-Trm that were sorted at low numbers.

The reviewer is right, there are proportionally less non-T_{RM} than T_{RM} in cervical tissues, which together with the limitation on the number of cells obtained for cervical tissue from patients, translates into very low number of non-T_{RM} cells sorted for this assay. This was clearly stated in the original paper, which shows all the sorted cells for each patient and acknowledges this limitation. Still, in the only sample where higher numbers of CD4⁺ non-T_{RM} were sorted (#M04, Table S1), only the CD4⁺ T_{RM} fraction was positive. Further, in the only sample where proviral DNA in the non-T_{RM} fraction was detected, there were >3 times more copies per cell in the T_{RM} fraction than in the non-T_{RM} fraction (5,073 copies/10⁶ vs 1,606 copies/10⁶). The scope of the paper was to assess if T_{RM} are an HIV reservoir, since this phenotype *per se* has never been addressed. While we confirm that T_{RM} are indeed an HIV reservoir in the cervical tissue, we do not exclude that non-T_{RM} cells can contribute to the HIV reservoir although, at least in mucosal tissues, where T_{RM} are predominant, their contribution would be marginal.

8. Why in Fig. 4B are the data presented as total CD4+ T cells vs. CD4+ Trm, when the sorted cells were CD4+ Trm vs. CD4+ non-Trm. Is it because most in almost all the non-Trm samples no HIV DNA was detectable?

That is exactly the reason. Only one sample, as mentioned in the paper, was positive in the non-T_{RM} fraction, and although we could show the value corresponding to the limit of detection, we preferred not to confuse the readers with some values calculated based on a very low frequent subset that could bias the results (as discussed in the previous question).

9. Fig. 5 describing the IHC experiments lack negative controls (e.g., staining from uninfected individuals, IHC controls, etc). It is not clear why in Fig. 5B the two representative samples (one on the left and one on the right) look so different. The one on the left looks like there is minimal CD69 staining, while the one on the right it looks like every cell has CD69 (background?) staining.

We completely agree on the importance of performing several controls to discard background and overall false detection rate. Indeed, we performed in situ hybridization of HIV-RNA in combination with CD69 antibody staining in two uninfected donors, with a mean of total positive cells of 0.03 cells/mm². We consider that this value corresponds to background from unspecific probes hybridization, since no positive cells were observed in controls without probes. In contrast, the mean frequency of total HIV-RNA+ cells in ART-suppressed patients was of 0.11 cells/mm², which was similar to what was observed in the cervical tissue from the viremic controller (with low viral load). We highlight that all patient samples performed

presented values above this background level and, importantly no positive cells for HIV-RNA were observed in the epithelium of uninfected donors, which reinforce the existence of T_{RM} harboring HIV-RNA⁺.

On the other hand, in Figure 6b we show two images with an HIV-RNA positive cell, one from the epithelium (left) and one from the submucosa (right). While in the image on the right we specifically selected an HIV-RNA positive cell within a CD69-enriched area, compatible with previously described T_{RM} areas or memory lymphocyte clusters, in the image on the left we show a positive cell from the epithelium. In the epithelium T_{RM} are embedded among epithelial cells as individual cells, infiltrated, without cluster formation. Thus, compared to a selected enriched area for CD69, there is overall less staining in the epithelium where all the epithelial cells are negative for this marker. Of note, controls without CD69 antibody were performed and no staining was observed, indicating the specificity of CD69 antibody.

All these information has now been clarified and included in the material and methods (lines 737-743).

10. The FMOs in Fig. S1 are not appropriate, as there is significant positivity in the FMO gates. There are no x- and y-axis labels.

As mentioned before, considering the opinion of all the reviewers and to comply with the journal requirements in terms of plot formatting we have modified all the plots in the manuscript (including axes scales, numbers and labels). Moreover, we have now performed new FMOs, since the small percentage shown in the FMOs of the old Fig.S1 corresponded to very few dots in samples with low numbers of cell events. These new FMOs do not modify the gating, which was already stringent for each of the molecules addressed. Yet, for readability and figure distribution reasons, in the new manuscript we only include the CD32/CD69 FMOs, which fit better in the new Fig.S1 covering specificity of CD32⁺ CD69⁺ cervical cells.

11. Fig. S2A is lacking the important control of an uninfected culture taken through the same time course.

We agree with the reviewer and we have now performed a time course of an additional tissue with and without HIV infection (same tissue). This experiment translated into 8 conditions, with 3, 5, 7 and 10 days of culture after HIV infection or mock-infection. Importantly, the dynamics observed for CD69 were similar for the two conditions, which did not modify the meaning of the results discussed in the original manuscript. This new experiment has now been included and corresponds to the new Fig. S3a. The corresponding results section has now been updated accordingly (lines 227-229).

Reviewer #2 (Remarks to the Author):

Summary. In this paper, the authors aimed to study the HIV reservoir in cervical tissues. A major focus was to understand if tissue resident memory (TRM) CD4⁺ T cells are infected with HIV and represent a sizeable proportion of the reservoir. The authors found that the

majority of CD4⁺ T cells in cervix expressed CD69, a canonical marker of TRMs, and demonstrate that many molecules associated with HIV susceptibility is expressed by TRMs and CD32⁺ TRMs. Quite remarkably, the authors found >200 times more HIV-proviral copies in cervical than blood CD4⁺ T cells. A vast majority of the total infected CD4⁺ T cells demonstrated a TRM phenotype.

This is an interesting paper for the HIV field and highlight that the reservoir composition in tissues is remarkable different from blood. It is the first data directly examining if TRMs are primary targets of HIV infection and viral persistence. HIV likely persist to a very high degree in TRMs given that the vast majority of lentiviral-infected cells are present in mucosal tissues (Estes et al, 2018, Nat Med). Given that a recent publication in mice found that CD4⁺ T cells in the female reproductive tract are essentially all TRMs (Beura et al, 2019, JEM; this publication should be cited), it altogether would make sense that TRMs are dominating in cervix also in humans. However, there are a couple of more experiments that could be done in order to truly validate this finding.

Major comments:

CD69 is a valid marker of TRMs, but not all CD69⁺ cells might be bona fide TRMs. In light of recent data from the Masopust lab and others (Beura et al, 2018, Immunity etc), it seems rationale to believe that CD62L⁻/CCR7⁻S1PR1⁻ CD69⁺ CD4⁺ T cells are the true TRMs in tissues. The authors do provide data that most CD69⁺ CD4⁺ T cells do not express CCR7 in the first supplementary figure, but also other egress receptors (such as S1PR1) and transcription factors (KLF2, KLF3 etc) are usually low in TRMs. Furthermore, TRMs usually have a specific gene signature that includes upregulation of specific chemokine receptors and transcription factors (for instance Hobit, Blimp-1 etc). Have the authors conducted RNAseq analysis or gene-expression analysis to confirm that the CD69⁺ CD4⁺ T cell signature in cervix is similar to a bona fide TRM signature published by others (for instance Kumar et al, 2017, Cell Reports)?

We agree with the reviewer on the importance of verifying the true T_{RM} nature of cervical CD69⁺CD4⁺ T cells in this paper, considering the lack of data in this respect. In performing the extensive phenotyping that we did for the original Figure 1, we have already confirmed the expression of certain markers associated to T_{RM}, as mentioned in the paper. Still, while we do not have data on the gene expression profile of CD69⁺ vs CD69⁻ CD4⁺ T cells from these tissues, we have performed extra analyses with a new panel of transcriptional factors and proteins associated to this phenotype in other human tissues or in mice (Beura et al. *J Exp Med*, 2018; Kumar et al. *Cell Reports*, 2017; and Topham et al. *Frontiers Immunol*, 2018). We now include these results in the new Figure 1, in which we show the frequencies of Eomes, T-bet, Hobit, S1PR1, CCR7, CD49a and PD-1 in cervical CD69⁻ and CD69⁺ CD4⁺ T cells. In agreement with previous reports, most of the CD69⁺ CD4⁺ T cells have low or absent expression of Eomes, T-bet, S1PR1 as well as of CCR7, and high expression of CD49a and PD-1. These results have now been included as a new section in the results (lines 78-99).

The authors treat infected blocks with CCL19 and CCL21 to assess the degree of infected cells in supernatant. Why did the authors try to use these chemokines given

that CCR7 is low on both CD69- and CD69+ CD4+ T cells in the cervix? The assay does not seem to work given that the number of infected cells is similar between the CCL19/CCL21 treated and untreated condition. The authors should if anything instead then use S1P in these assays to rule out S1PR1+ T cells – which should be the CD69- CD4+ T cells.

We agree with the reviewer and we have performed a new experiment including S1P (*see following response*).

Furthermore, despite more cells in the supernatant are CD69-, there is also a heavy skewing of infected cells being more CD69+ in the supernatant. Based on these data, it is not possible then to conclude that migrating infected CD4+ T cells have a circulating CD69- phenotype. The authors could instead try a trans well migration assay and supplement with CCL19, CCL21 and S1P to determine if infected cells are moving towards a gradient in the lower wells. This is the most established way in the field to provide some evidence of migration capacity.

We thank the reviewer for this suggestion and we have now modified the experiment accordingly. We have performed this assay with a transwell plate to demonstrate simultaneous migration towards CCL19, CCL21 and S1P in the lower wells. The concentration and details for this assay are now specified in the material and methods section (lines 679-687). The addition of these molecules marginally increased infection in the lower-well supernatants (with an increase of about 3% over the not-treated infected conditions in the supernatant). Importantly, substantial infection was confined to the tissue blocks (19%), which was not modified by the addition of the attracting molecules. Moreover, the infection confined to the tissue was remarkably associated to CD69⁺/CD69⁺ CD103⁺ expression. These results and the new Figure S3c have now been included in the revised manuscript (lines 237-248).

[redacted]

Why do the authors only gate for CD32dim cells in their data set? The initial Nature paper also included CD32hi cells.

We have extensively covered this concern on CD32 expression in the first response to Reviewer 1. To properly answer this reviewer, we excluded CD32high cells because later papers (such as *Pérez et al. Nature, 2018* and *Osuna et al. Nature, 2018*) demonstrated contamination with B cell/CD14⁺ conjugates in this fraction. Further, we have made an effort to better show the flow of the gating strategies, the purity of these cells, as well as the reasoning behind the CD32 analyses and the conclusions obtained. This information and data has now been included in the material and methods section (lines 635-643), results (lines 109-125) and discussion (lines 470-493).

In the field there is a debate on whether HIV replication is ongoing or not after ART. The authors found >200 times more proviral DNA copies per cell in cervix in comparison to blood – this is a remarkable number. Despite that, the authors claim that few TRMs are activated. There are literature suggesting that TRMs are in a pseudo-activated state and poised to rapidly exert effector functions following antigen recognition. The authors also demonstrate more HLA-DR+ cells within the CD4 compartment in HIV+ subjects despite ART, which could suggest that some cells are activated and ongoing viral replication might take place despite ART.

Is there any evidence of more proviral copies within the CD69+HLA-DR+ compartment in cervix?

[redacted] Could it be poor penetration of drugs in the cervix explaining the higher HIV-DNA levels in this site, similar to lymphoid tissues?

The referee makes very interesting reflections regarding the association between HLA-DR expression, activation and potential ongoing HIV replication, and we agree with this line of thought. Unfortunately, we do not have data on the proviral DNA by CD69⁺DR⁺ or CD69⁺DR⁻, yet in the original manuscript we already showed that, in the explant model, p24⁺T_{RM} cells express more HLA-DR than p24⁻T_{RM} (new Fig. 3e).

[redacted]

Regarding the last comment, yes, it could be poor penetration, as a high degree of variability in penetration of the different drugs has been observed in the female genital tract (*Thompson et al. J Acquir Immune Defic Syndr, 2013*). We have added a comment referring to this phenomenon in the discussion (lines 588-591).

Minor comments:

Overall, the authors need to reformat many of the flow plots. It seems like many flow plots have been directly transported from flowjo and not really formatted in illustrator or another program to remove unnecessary gating data. Be consistent at least and label the X- and Y-axis with the markers and fluorophores in the same way for all plots and remove data within the gates except for the percentage. Make the X- and Y-legends for each marker larger (similar to Figure S2) and keep the labeling consistent. Avoid overlap of multiple gates over each other if possible as it becomes hard to visualize the data.

We apologize for the lack of consistency on the flow plots. We have now modified all corresponding figures to improve the understanding of the flow and consistency on the esthetics, but also complying with the journal requirements in terms of plot formatting (i.e. including axes scales and numbers). We hope that the new Figure formatting is now clearer and more attractive visually.

Reviewers' comments:

Reviewer #1 (Remarks to the Author):

This manuscript is much improved from the original version, and the FACS data are much more interpretable in the revised figures. I feel however that there are some issues that still need to be addressed, as detailed below.

1. To provide direct evidence that the CD69+ cells are more susceptible to ex vivo infection by HIV compared to the non-Trm CD69- cells, the authors sorted these populations and then infected them ex vivo with HIV. This is indeed the most direct way to demonstrate this point, and this key experiment is presented in Figure S3b. From the figure, it is apparent that many more Trm were sorted out than the non-Trm, consistent with Trm being much more abundant. This is, however, a concern when the authors compare the % of p24+ cells. The FACS plots in Fig. S3b show that 9.5% of the cells for the Trm were p24+, while 0% were for the non-Trm. However, it's possible that no cells fell within the p24+ gate simply because there were too few total cells analyzed. To account for this, the authors should reanalyze the data by down-sampling the Trm data to match the total number of cells analyzed to that of the corresponding non-Trm. Also, the FACS plots of p24 positivity for the Trm and non-Trm for all four of the donors shown in Fig. S3b should be presented.

2. The authors state that they did not detect any enhancement of CD69 expression in the sorted CD69- population after infection (lines 254-255 in marked up version), supporting the notion that CD69 is not upregulated during HIV infection. However, these data were not shown anywhere and should be a part of Figure S3.

3. There are comments throughout the manuscript that CD32+ cells are preferentially infected. (e.g., Lines 279-280 that "CD32+ Trm preferentially support HIV infection ex vivo", Lines 513-514 that "CD32 expression in CD4+ Trm identifies a subset remarkably susceptible to HIV-1 infection", etc). Without an experiment equivalent to that of the CD69 pre-sorting experiment, one cannot rule out that CD32 is higher on infected cells simply due to HIV-mediated upregulation of CD32. This is important because CD32 has been reported to be upregulated by HIV during infection. It is fine to state that infected cells have higher levels of CD32, but there are no data presented by the authors that it's not due to HIV-induced upregulation of CD32.

4. Figure 6. In response to the comment that negative controls were missing, the authors explained that negative controls of samples from uninfected individuals and controls without the CD69 antibody were performed, and had now added these explanations to the Methods. These controls need to not just be explained in the Methods, but also shown, if not in the main figure then as a supplementary figure. The authors commented that in uninfected donors, they observed HIV-RNA in 0.03 cells/mm², while in infected individuals it was observed at 0.11 cells/mm². Because this is only a difference of 3.7-fold, it is important to show how data were adjusted for background. E.g., could the authors demonstrate that in uninfected individuals, the HIV-RNA signal was not more

abundant in the CD69+ cells? Because if it was, it questions the specificity of the observation in the infected individuals.

5. [redacted]

Reviewer #2 (Remarks to the Author):

The authors have answered all my questions.

Reviewer #1 (Remarks to the Author):

This manuscript is much improved from the original version, and the FACS data are much more interpretable in the revised figures. I feel however that there are some issues that still need to be addressed, as detailed below.

1. To provide direct evidence that the CD69+ cells are more susceptible to ex vivo infection by HIV compared to the non-Trm CD69- cells, the authors sorted these populations and then infected them ex vivo with HIV. This is indeed the most direct way to demonstrate this point, and this key experiment is presented in Figure S3b. From the figure, it is apparent that many more Trm were sorted out than the non-Trm, consistent with Trm being much more abundant. This is, however, a concern when the authors compare the % of p24+ cells. The FACS plots in Fig. S3b show that 9.5% of the cells for the Trm were p24+, while 0% were for the non-Trm. However, it's possible that no cells fell within the p24+ gate simply because there were too few total cells analyzed. To account for this, the authors should reanalyze the data by down-sampling the Trm data to match the total number of cells analyzed to that of the corresponding non-Trm. Also, the FACS plots of p24 positivity for the Trm and non-Trm for all four of the donors shown in Fig. S3b should be presented.

The reviewer is right regarding the limitation of the comparison between the two subsets, due to a significant difference in sorted total numbers. We have now analyzed the data as requested. If we match samples for the same number of TRM as of non-TRM, three out of four samples should have shown some positive cell, while one tissue had a chance of 0.83 to show one. Summarizing these data, the number of p24+ events expected to be detected in each subset, based on the number of non-TRM events, follows:

	TRM	non-TRM
1	3.12	2
2	0.83	0
3	4.44	0
4	1.4	0

These values correspond to a 1.98-fold change in CD69+ vs CD69-, which agrees with the values reported by Joag et al. (Mucosal Immunology, 2016). These authors, as already referenced in the manuscript, demonstrated a preference for HIV towards CD69+CD4+T cells in the cervix, where HIV entry was increased by 1.9-fold (Figure below) in comparison to CD69-.

We have included another example of these plots, which now shows donors #3 and #4 in Figure S3. However, we prefer not to include donors #1 and #2 because CD69 expression was labeled with a different fluorochrome, FiTC, of the same clone (FN50) than the one used for sorting before infection (PE-CF594). So in those experiments, the TRM fraction after infection may not show the real level of CD69 expression (evaluated in the FiTC) because of the pre-labeling with PE-CF594. Instead, in experiments #3 and #4, we used the same FiTC anti-CD69 to label these cells for sorting and for p24 evaluation, 3 days after infection. We prefer to show all the plots only to the reviewer and the editor for this reason, although the results are not affected, it would require further explanations to the readers. Importantly, as a requirement of the journal guidelines, all the row data will be supplied as an excel data sheet and, thus, readers will be able to check for themselves the actual values of all the experiments.

2. The authors state that they did not detect any enhancement of CD69 expression in the sorted CD69- population after infection (lines 254-255 in marked up version), supporting the notion that CD69 is not upregulated during HIV infection. However, these data were not shown anywhere and should be a part of Figure S3.

The examples in the plots presented in Fig. S3b show p24 in the X axes and CD69 in the Y axes. In these examples one can see that while TRM express >70% of CD69, less than 12% of non-TRM upregulate CD69 and none of them were infected. As we have explained in the previous point, while we include the data here for the reviewer's evaluation, we rather show only the two examples performed with the same clone to not confuse the readers. Importantly, CD69 was minimally upregulated in non-TRM from any of the donors.

3. There are comments throughout the manuscript that CD32+ cells are preferentially infected. (e.g., Lines 279-280 that "CD32+ Trm preferentially support HIV infection ex vivo", Lines 513-514 that "CD32 expression in CD4+ Trm identifies a subset remarkably susceptible to HIV-1 infection", etc). Without an experiment equivalent to that of the CD69 pre-sorting experiment, one cannot rule out that CD32 is higher on infected cells simply due to HIV-

mediated upregulation of CD32. This is important because CD32 has been reported to be upregulated by HIV during infection. It is fine to state that infected cells have higher levels of CD32, but there are no data presented by the authors that it's not due to HIV-induced upregulation of CD32.

We agree with the reviewer and we have modified these statements to not confuse the readers. We now stress the fact that we cannot be sure of this subset's increased permission to HIV. Unfortunately, it would be extremely difficult to obtain enough cervical cells from sorting to infect them afterwards and assess p24 infection at day 3 in the TRM CD32+ fraction vs the negative (and include all the necessary uninfected controls). Thus, we have tried to clearly acknowledge this limitation rephrasing those lines (lines 253 and 477).

4. Figure 6. In response to the comment that negative controls were missing, the authors explained that negative controls of samples from uninfected individuals and controls without the CD69 antibody were performed and had now added these explanations to the Methods. These controls need to not just be explained in the Methods, but also shown, if not in the main figure then as a supplementary figure. The authors commented that in uninfected donors, they observed HIV-RNA in 0.03 cells/mm², while in infected individuals it was observed at 0.11 cells/mm². Because this is only a difference of 3.7-fold, it is important to show how data were adjusted for background. E.g., could the authors demonstrate that in uninfected individuals, the HIV-RNA signal was not more abundant in the CD69+ cells? Because if it was, it questions the specificity of the observation in the infected individuals.

Two images of each of the different controls performed for the ISH-IHC analyses are now included as Supplementary Figure 5. Top images show images of the cervical epithelium and the lamina propia of an uninfected patient, middle images show close images of false positive cells in the cervical tissue of these uninfected patients and, lastly, bottom images show the IHC control with no labeling against CD69 (secondary Ab only) neither against vRNA. Regarding the reviewer's question on the HIV-RNA signal of the uninfected women, indeed, these false positive cells were absent from the CD69+ regions mentioned in the case of the infected patients, as well as never present in the epithelium. These facts have been emphasized in the manuscript now (lines 732-735).

[redacted]

REVIEWERS' COMMENTS:

Reviewer #1 (Remarks to the Author):

They have addressed point #1 by re-analyzing and showing more of the data with the CD69+ cells, and modified the manuscript to not make the unsubstantiated claim CD32+ cells are being more susceptible since CD32 is known to be upregulated by HIV infection. [redacted]